# The steroid-hormone ecdysone coordinates parallel pupariation neuromotor and morphogenetic subprograms via epidermis-to-neuron Dilp8-Lgr3 signal induction

Fabiana Heredia[1,12], Yanel Volonté[1,2,12], Joana Pereirinha [1,6,12], Magdalena Fernandez-Acosta[1], Andreia P. Casimiro[1], Cláudia G. Belém[1,7], Filipe Viegas [1,8], Kohtaro Tanaka[3,9], Juliane Menezes[1], Maite Arana[2], Gisele A. Cardoso[1,4,10], André Macedo [1], Malwina Kotowicz [1,11], Facundo H. Prado Spalm[2], Marcos J. Dibo [2], Raquel D. Monfardini[1,4], Tatiana T. Torres [4], César S. Mendes[1], Andres Garelli [1,2 ✉] & Alisson M. Gontijo [1,5 ✉]

Innate behaviors consist of a succession of genetically-hardwired motor and physiological subprograms that can be coupled to drastic morphogenetic changes. How these integrative responses are orchestrated is not completely understood. Here, we provide insight into these mechanisms by studying pupariation, a multi-step innate behavior of *Drosophila* larvae that is critical for survival during metamorphosis. We find that the steroid-hormone ecdysone triggers parallel pupariation neuromotor and morphogenetic subprograms, which include the induction of the relaxin-peptide hormone, Dilp8, in the epidermis. Dilp8 acts on six Lgr3-positive thoracic interneurons to couple both subprograms in time and to instruct neuro-motor subprogram switching during behavior. Our work reveals that interorgan feedback gates progression between subunits of an innate behavior and points to an ancestral neuromodulatory function of relaxin signaling.

[1] CEDOC, Chronic Diseases Research Center, NOVA Medical School | Faculdade de Ciências Médicas, Universidade Nova de Lisboa, Lisbon, Portugal. [2] INIBIBB, Instituto de Investigaciones Bioquímicas de Bahia Blanca, Universidad Nacional del Sur - CONICET, Bahía Blanca, Argentina. [3] Instituto Gulbenkian de Ciências, Oeiras, Portugal. [4] Laboratório de Genômica e Evolução de Artrópodes, Departamento de Genética e Biologia Evolutiva, Universidade de São Paulo, São Paulo, Brazil. [5] The Discoveries Centre for Regenerative and Precision Medicine, Lisbon Campus, Rua do Instituto Bacteriológico 5, 1150-190, Lisbon, Portugal. [6] Present address: Institute of Molecular Biology, Mainz, Germany. [7] Present address: The Francis Crick Institute, London, UK. [8] Present address: Department of Molecular Life Sciences, University of Zurich, Zurich, Switzerland. [9] Present address: Fred Hutchinson Cancer Research Center, Seattle, WA, USA. [10] Present address: CBMEG, Universidade Estadual de Campinas, Campinas, Brazil. [11] Present address: DZNE, Helmholtz Association, Bonn, Germany. [12] These authors contributed equally: Fabiana Heredia, Yanel Volonté, Joana Pereirinha. ✉email: agarelli@inibibb-conicet.gob.ar; alisson.gontijo@nms.unl.pt

nnate (i.e., intrinsic) behaviors are genetically-hardwired behaviors that do not require previous learning or experience for proper execution[1,2]. These behaviors comprise neuro-motor and physiological subprograms that are many times coupled to drastic morphogenetic changes. For instance, males of some pacific salmonid species undergo dramatic morphological changes during spawning migration[3], and females of many mammalian species remodel their pubic ligaments to favor delivery during parturition behavior[4–6]. Holometabolan insects, those with complete metamorphosis, have evolved different innate behaviors and processes that promote survival during this critical life-history stage[7–9]. Honey bees, for instance, metamorphose within wax-sealed hive chambers, while some lepidopterans (butterflies and moths) pupate within spun cocoons. Cyclorraphous flies, such as *Drosophila*, undergo metamorphosis within a puparium, a protective capsule consisting of the reshaped and hardened ecdysed cuticle of the last larval instar[7,8,10].

Puparium formation (pupariation) is associated with additional survival-promoting behaviors, including the extrusion of anterior spiracles for efficient gas exchange, and the expulsion and spreading of a salivary-gland-derived secretory "glue" that attaches the puparium to its substrate[11–14]. Proper pupariation therefore requires tight coordination between associated behavioral subprograms, body-reshaping motor subprograms, and non-motor morphogenetic processes, such as the cuticle sclerotization subprogram that fixes the achieved morphological changes of the puparium[7,10,15]. Although the whole pupariation process is known to be triggered by a surge in the levels of the steroid hormone 20-hydroxyecdysone (20HE) at the end of the third instar larval phase[7,16–18], the downstream programs are thought to be mediated by specific neuroendocrine signals and circuits, most of which remain to be characterized[7,19–22]. Furthermore, how subprograms of innate behaviors are coordinated amongst themselves and in time is not fully understood.

During the first half of the last larval instar, an imaginal disc growth checkpoint system mediated by the disc-derived stress signal, the relaxin-like peptide hormone *Drosophila* insulin-like peptide 8 (Dilp8)[23,24], and its neuronal receptor, the Leucine-Rich Repeat containing G protein-coupled receptor 3 (Lgr3)[25–28], contributes to the development of proportionate adult body parts by transiently antagonizing 20HE biosynthesis by the prothoracic gland[23–46].

Here, we show that coordination between early behavioral, neuromotor, and morphogenetic subprograms of pupariation requires the 20HE-dependent induction of Dilp8-Lgr3 signaling from the cuticle epidermis to the central nervous system at the onset of pupariation. This interorgan signaling event is critical for proper progression of the pupariation motor program.

## Results

**The Dilp8-Lgr3 pathway is required for puparium morphogenesis.** We serendipitously found that mutation (*Lgr3^ag1*)[26] or ubiquitous RNA interference (RNAi)-mediated knockdown of *Lgr3* using the GAL4-UAS system (*UAS-Lgr3-IR*)[26], generates aberrantly shaped puparia that are slightly thinner and more elongated than their wild-type (WT) background controls, as measured by puparium aspect ratio (AR = length/width) (Fig. 1a–d). Less penetrant phenotypes include defective retraction of the anteriormost pre-spiracular segments into the body (anterior retraction) and failure to extrude the anterior spiracles. Similar phenotypes were observed in five *dilp8* loss-of-function mutants generated here by CRISPR/Cas9-mediated directed mutagenesis[47,48] (Fig. 1e, f, Supplementary Fig. 1a),

upon ubiquitous RNAi knockdown of *dilp8* (*dilp8-IR^TRIP*, see also Methods and Supplementary Fig. 1b–g), and in an independent knock-out allele *dilp8^KO* (ref. [40], and Supplementary Fig. 1h, i). These findings suggested that animals lacking Dilp8-Lgr3 signaling have problems contracting their body into the puparium shape and/or stabilizing their remodeled body at the contracted state.

**Lgr3 is required in a subpopulation of neurons for proper puparium morphogenesis.** To ask in which tissue *Lgr3* is required for puparium morphogenesis control, we carried out tissue-specific *Lgr3* RNAi knockdown. Puparium AR was most strongly increased when *Lgr3* was knocked-down in neurons using the pan-neuronal synaptobrevin promoter-fusion GAL4 line, *R57C10-GAL4* (*R57C1>*)[49,50] (Fig. 1g). As *Lgr3* is also required in neurons for imaginal disc growth coordination before the midthird instar transition[23–28,34,46] (Fig. 1h, i), this finding poses the question if the puparium morphogenesis defect of *dilp8* and *Lgr3* mutants arises from the abrogation of this same early signaling event.

Imaginal disc-derived Dilp8 acts on a subpopulation of *Lgr3*-positive CNS neurons that can be genetically manipulated using the cis-regulatory module *R19B09*[25–28] (Fig. 1h–i and Supplementary Fig. 1j), which consists of the ~3.6-kb 7th intron of the *Lgr3* locus[49,51,52]. R19B09-positive cells include a bilateral pair of neurons, the *pars intercerebralis Lgr3*-positive (PIL)/growth coordinating *Lgr3* (GCL) neurons, which respond to Dilp8 by increasing cAMP levels, and are thus considered the major candidate neurons to sense the Dilp8 imaginal tissue growth signal[25–27,46]. We reasoned that if the neurons that require *Lgr3* to inhibit ecdysone biosynthesis upon imaginal tissue stress are the same neurons that require *Lgr3* to control puparium morphogenesis, then knockdown of *Lgr3* in *R19B09*-positive cells, but not in the other *Lgr3* cis-regulatory module-positive cells, should increase puparium AR. Accordingly, RNAi knockdown of *Lgr3* using *R19B09-GAL4* (*R19B09 > Lgr3-IR*), but not four other *Lgr3* cis-regulatory module GAL4 lines tested (Fig. 1i), specifically suppresses the developmental delay caused by *dilp8* overexpression under the direct control of the ubiquitous *tubulin* (*tub*) promoter [*tub-dilp8*; ref. [27]] (Fig. 1j). However, *R19B09 > Lgr3-IR* had no effect on puparium AR (Fig. 1k). Of the four other cis-regulatory-module-GAL4 lines tested, only *R18A01-GAL4* (*R18A01 >*, Supplementary Fig. 1k) strongly increased puparium AR when driving *Lgr3-IR* (Fig. 1i, k). These results clearly show that proper puparium morphogenesis does not require Lgr3 in the growth-coordinating R19B09 neurons. Instead, Lgr3 is required in a different population of cells that can be genetically manipulated using the *R18A01 >* driver. Consistent with this, the puparium morphogenesis defect of *Lgr3^ag1* mutants could be completely rescued by expressing an *Lgr3* cDNA [*UAS-Lgr3*; ref. [26]] under the control of *R18A01 >* (Fig. 1l). We conclude that the control of the onset of metamorphosis and the control of puparium morphogenesis are two independent processes that require *Lgr3* in two separate populations of neurons marked by *R19B09 >* and *R18A01 >*, respectively (Fig. 1i, Supplementary Fig. 1j, k).

**20HE signaling induces *dilp8* transcription in the cuticle epidermis during pupariation.** We next investigated the source of the Dilp8 signal that controlled puparium morphogenesis. A series of genomewide transcriptional studies indicated that *dilp8* transcripts are strongly upregulated in the "carcass," a tissue composed majorly of cuticle epidermis and muscle, and to a lesser extent of sessile hemocytes, neurons, and other cell types, at the onset of pupariation [white prepupae (WPP T0)], and in

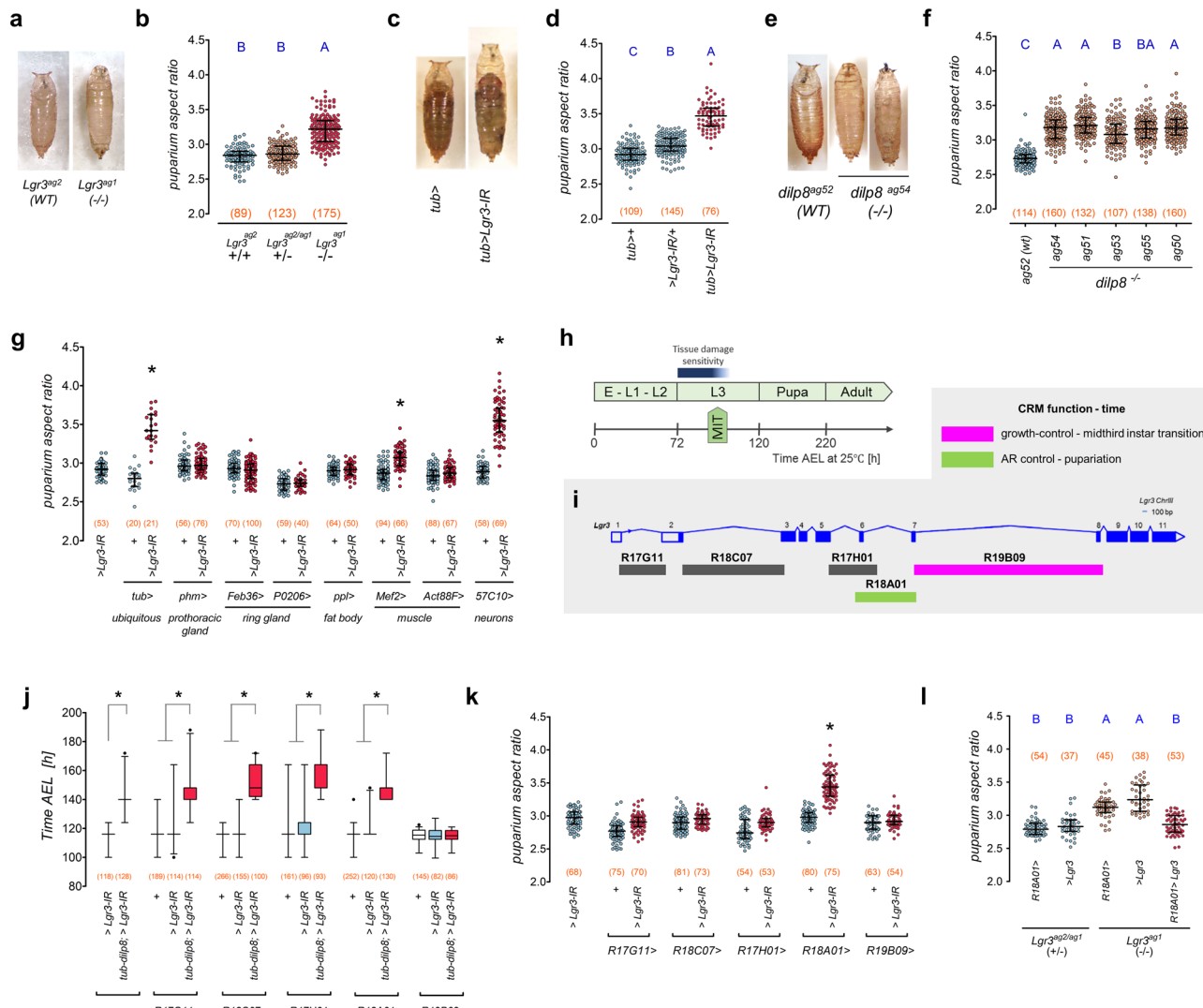

**Fig. 1 Puparium morphogenesis requires Dilp8-Lgr3 signaling in neurons. a** Representative photos of puparia from the depicted genotypes. **b** *Lgr3* mutation increases puparium aspect ratio (AR). Shown are dot plots of puparium AR. **c** Representative photos of puparia from the depicted genotypes. **d** Ubiquitous *Lgr3* knockdown with *tubulin-GAL4* (*tub >*) (*tub > Lgr3-IR*) increases puparium AR. Shown are dot plots of puparium AR. **e** Representative photos of puparia from the depicted genotypes. **f** *dilp8* mutation increases puparium AR. Shown are dot plots of puparium AR. **g** Pan-neuronal *Lgr3* knockdown (*57C10 >*) increases puparium AR similarly to ubiquitous knockdown (*tub >*). Shown are dot plots of puparium AR. **h** Sensitivity to tissue-damage-induced Dilp8 occurs before the midthird instar transition (MIT). Time after egg laying (AEL). **i** *Lgr3* locus scheme with its cis-regulatory modules (CRM) and known activities. **j** *tub-dilp8*-induced developmental delay rescue by *R19B09 > Lgr3-IR*. Box plots showing pupariation time. **k** Knockdown of *Lgr3* in R18A01, but not in R19B09 neurons, increases the puparium AR. Shown are dot plots of puparium AR. **l** Rescue of the puparium AR defect of *Lgr3ag1* mutants by *R18A01 > Lgr3*. Shown are dot plots of puparium AR. Statistics (full details in Supplementary Table 2): **b, d, f, g, k, l** Dots: one animal. Horizontal bar, median. Error bars: 25–75% percentiles. **j** Box, 25–75%; horizontal bar, median; whiskers, 5-95%. Dots, outliers. **b, d, f, l** Same blue letter, *P* > 0.05. **b, d, f, l** Dunn's test. **g, j, k** Dunn's test, compared to both *>Lgr3-IR* and respective *GAL4 > +* control. (N) Number of animals (orange). **P* < 0.05.

an ecdysone-receptor-dependent manner[46,53–55] (Supplementary Fig. 2a–c). We confirmed and expanded these data using quantitative reverse-transcriptase polymerase chain reaction (qRT-PCR) with cDNA obtained from whole synchronized larvae or their dissected tissues (see Methods). We find that *dilp8* mRNA levels increase three orders of magnitude between post-feeding 3rd instar larvae (i.e., "wandering" stage) and early pupariating animals, and a decrease in the peak can be detected as soon as 1.0 h after WPP T0 ("T60," Fig. 2a). The WPP T0 upregulation is largely explained by a strong increase in carcass-derived *dilp8* mRNA (Fig. 2b). These results suggest that the upregulation of *dilp8* mRNA in the carcass at pupariation is part of the normal developmentally-triggered 20HE-dependent pupariation process[17].

However, as the whole pupariation program is dependent on 20HE signaling, which peaks −4 h before pupariation[17,54], it is not clear whether or not this is a consequence of direct action of 20HE on the carcass. To test this, we incubated dissected carcasses from 3rd instar larvae collected at 96 h after egg laying with 20HE for 3 or 6 h and assayed for *dilp8* mRNA levels by qRT-PCR. As expected, *dilp8* mRNA was upregulated by a factor of 30.0 at 6 h and was not affected by ethanol vehicle treatment (Fig. 2c), indicating that *dilp8* is a direct target of 20HE-dependent signaling in the carcass. The *dilp8* expression pattern was nevertheless different from another 20HE-dependent epidermis-transcribed gene, *pale* (*ple, Tyrosine 3-monooxygenase*)[54,55] (Supplementary Fig. 2d–f). While 20HE also slightly stimulated *pale* mRNA levels, the carcass cells obtained 96 h after egg laying were clearly already

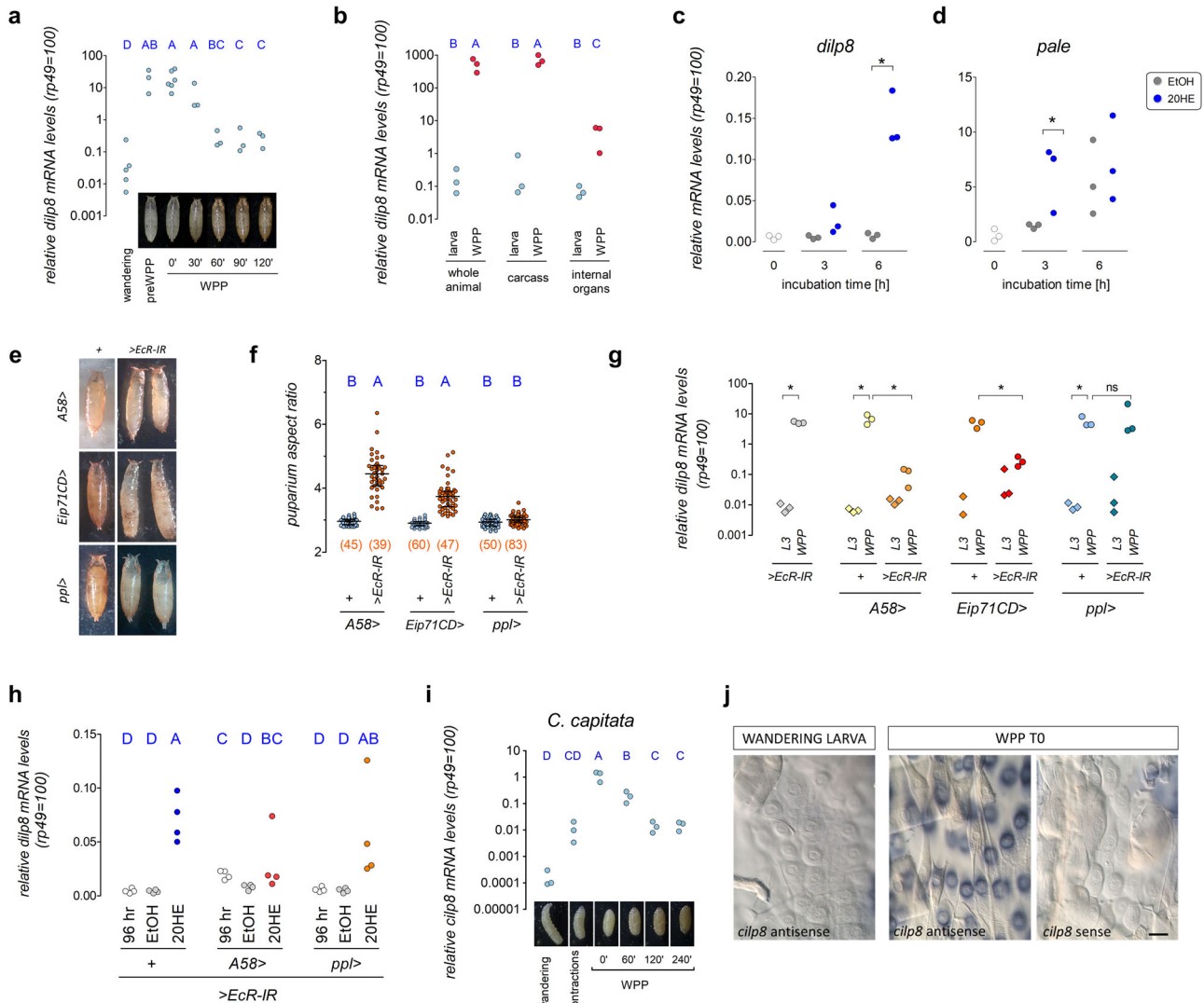

**Fig. 2 Ecdysone induces a conserved *dilp8* expression peak in the cuticle epidermis during pupariation. a** *dilp8* transcription peaks at pupariation. Shown are dot plots of qRT-PCR estimations of *dilp8* mRNA levels. **b** *dilp8* transcripts are enriched in the white-prepupa (WPP) carcass (integument and body wall muscles). Shown are dot plots of qRT-PCR estimations of *dilp8* mRNA levels. **c** *dilp8* and **d** *pale* mRNA levels in carcasses treated with 20HE or etOH in vitro. Shown are dot plots of qRT-PCR estimations of mRNA levels of each gene. **e** Representative photos of puparia from the depicted genotypes. **f** Knockdown of *EcR* in epidermal cells with *A58 >* and *Eip71CD >*, but not in fat body with *ppl >*, increases puparium aspect ratio (AR). Shown are dot plots of puparium AR. **g** Knockdown of *EcR* in epidermal cells with *A58 >* and *Eip71CD >*, but not in fat body with *ppl >*, reduces *dilp8* mRNA expression at the WPP T0 stage. Shown are dot plots of qRT-PCR estimations of *dilp8* mRNA levels. **h** Knockdown of *EcR* in epidermal cells with *A58 >*, but not in fat body with *ppl >*, supresses 20HE-dependent *dilp8* transcription in isolated carcasses. Shown are dot plots of qRT-PCR estimations of *dilp8* mRNA levels. **i** *C. capitata ilp8* (*cilp8*) is transiently strongly expressed at WPP T0. Shown are dot plots of qRT-PCR estimations of *cilp8* mRNA levels. **j** In situ hybridization with *cilp8* antisense probes stains epidermal cells on *C. capitata* WPP T0 carcasses. Representative image of at least 3 animals per condition. Statistics (full details in Supplementary Table 2): **a, b, c, d, g-i** Dots: biological repeats **f** Dots: one animal. Horizontal bar, median. Error bars: 25-75% percentiles. **a, b, f, h, i** Same blue letter, *P* > 0.05. **a, b, h, i** Student-Newman-Keuls test. **f** Dunn's test. **g** Holm-Sidak test. **c, d** Student's *t*-test. *$P < 0.05$. (N) Number of animals (orange). Scale bar, 50 μm.

committed to increase *pale* mRNA levels, independently of further 20HE exposure (Fig. 2d). This is consistent with previous reports that show *pale* mRNA levels already upregulated at −4 h before pupariation, whereas *dilp8* mRNA levels remain at basal level[55] (Supplementary Fig. 2a, d). These results suggest that *dilp8* is a direct or indirect target of 20HE in the larval carcass. The timing of the *dilp8* transcriptional response to 20HE are consistent with a model where *dilp8* is a direct target of very late 20HE-dependent signaling, probably the strongest and last peak preceding pupariation (at −4 h), whereas *pale* is induced by smaller and earlier 20HE peaks, probably the midthird-instar transition peak, which is linked to the initiation of salivary glue protein production in the salivary gland[18,34].

To genetically test if *dilp8* transcription in the epidermis occurs downstream of 20HE signaling, we knocked-down the *ecdysone receptor* (*EcR*) gene with RNAi (*UAS-EcR-IR*) in the epidermis using two epidermal GAL4 lines *A58-GAL4* and *Eip71CD-GAL4* (*A58 > EcR-IR* and *Eip71CD > EcR-IR*; see Supplementary Fig. 2g) and quantified puparium AR and *dilp8* mRNA levels by qRT-PCR in synchronized wandering stage (108 h after egg laying) and WPP T0 stage animals. The *UAS-EcR-IR* transgene alone (*EcR-IR/ +*) and *EcR* knockdown in the fat body using the *pumpless-GAL4* line (*ppl > EcR-IR*) served as a negative controls for epidermal expression (Supplementary Fig. 2g). Results showed a statistically-significant increase in puparium AR in *A58 > EcR-IR* and *Eip71CD > EcR-IR* animals, but not in all other controls

(Fig. 2e, f). Furthermore, as expected, we observed a statistically-significant decrease in *dilp8* mRNA levels in *A58 > EcR-IR* and *Eip71CD > EcR-IR* WPP T0 animals, but not in all other controls (Fig. 2g). We conclude that epidermal *EcR*, but not fat body *EcR*, is critical for the achievement of peak *dilp8* mRNA levels in WPP T0 animals. Interestingly, the puparium AR increase produced by *EcR* knockdown in the epidermis was much stronger than what we observed in *dilp8* or *Lgr3* animals (compare Fig. 1a–f with Fig. 2e, f). This is consistent with a scenario where ecdysone signaling regulates additional aspects required for proper puparium morphogenesis, apart from *dilp8* transcription, such as cuticle sclerotization. Accordingly, the cuticle of *A58 > EcR-IR* animals partially or completely fails to sclerotize, whereas cuticle sclerotization is apparently complete in *dilp8* or *Lgr3* mutants). In line with this rationale, a fraction of the control *ppl > EcR-IR* animals had defective anterior retraction (Fig. 2e, lower panels), which could suggest a role for the fat body (or any other *ppl > -positive tissue) in this process. We nevertheless hypothesize this is unlikely to be related to the expression of *dilp8* in the epidermis, as *ppl > does not drive significant expression in cuticle epidermal cells at this developmental stage (Supplementary Fig. 2g).

The fact that *A58 > EcR-IR* and *Eip71CD > EcR-IR* WPP T0 animals are so severely affected and that the *dilp8* mRNA peak is so sharp in time, can cast doubt on the precision of the samples collected, despite our efforts to avoid this problem by carefully monitoring each animal and establishing criteria for WPP T0 as wandering behavior cessation, spiracle extrusion, and body contraction cessation. To circumvent this limitation, we dissected the carcass of staged, 96-h *A58 > EcR-IR* larvae and quantified *dilp8* mRNA levels following incubation with 20HE or vehicle for 6 h ex vivo, as performed above. Carcasses from control animals carrying the *EcR-IR* transgene alone or with *EcR* knockdown in the fat body (*ppl > EcR-IR*) served as controls. Results showed that whereas 20HE strongly induced *dilp8* in *EcR-IR/ +* or *ppl > EcR-IR* animals, there was no statistically-significant induction of *dilp8* by 20HE in the carcasses of *A58 > EcR-IR* animals (Fig. 2h). Even though we have not assayed for direct binding of EcR to the *dilp8* locus, the results described above are consistent with a cell-autonomous, direct regulation of *dilp8* by the *EcR*. Furthermore, we can conclude that 20HE activity upstream of *dilp8* during pupariation is the opposite of what occurs in early 3rd instar larvae, when Dilp8 originating from abnormally-growing imaginal discs acts upstream of 20HE, inhibiting its biosynthesis[23–28,34,46].

**The *ilp8* transcriptional peak at pupariation is conserved in a distant cyclorrhaphan.** We next asked if this *ilp8* peak at pupariation is conserved in other puparium-forming insects. For this, we characterized the pupariation program of the Tephritidae fly *Ceratitis capitata* (Fig. 2i; see Methods). We extracted mRNA from animals synchronized at specific stages of pupariation and quantified the *Ceratitis insulin-like peptide 8* ortholog (*cilp8*) mRNA levels using qRT-PCR and the *Ceratitis rp49* ortholog as a control gene. Our results show a very strong, up to four-orders of magnitude, upregulation of *cilp8* mRNA levels at WPP "T0" (Fig. 2i). Interestingly, the levels of *cilp8* mRNA were already upregulated by a factor of ~88 at the ~5-min "body contraction" phase that precedes early WPP formation by 1–1.5 h (Fig. 2i), suggesting that *cilp8* can act very early or before the pupariation behavior begins. The levels at 2 h after T0 (T120) were still ~100-fold higher than wandering stage larvae (Fig. 2i), indicating that the *ilp8* peak might be broader in *C. capitata* than in *D. melanogaster*. Nevertheless, these results indicate that the upregulation of *ilp8* at the time of puparium formation has been conserved for at least the time since *Drosophila* and *Ceratitis* shared their last

common ancestor ~126 million years ago (MYA) [confidence interval (97-153 MYA)][56].

To pinpoint the source of *cilp8* upregulation in the carcass of WPP T0 animals, we carried out in situ hybridization using a *cilp8* antisense probe. Strong staining was detected in epidermal cells of the cuticle of WPP T0 animals (Fig. 2i). Consistently, no signal was detectable in post-feeding 3rd instar larvae or in WPP T0 animals probed with a control sense *cilp8* probe (Fig. 2j). These results corroborate the findings in *Drosophila*, strongly suggesting that a conserved surge of *ilp8* occurs in the cuticle epidermis downstream of the 20HE signaling event that instructs the animal to initiate the pupariation program.

**Dilp8 is required during pupariation for proper puparium morphogenesis.** To genetically test if the pupariation-associated *dilp8*-mRNA peak is the primary source of Dilp8 activity that signals to Lgr3 in *R18A01* neurons to mediate proper puparium morphogenesis, we hypothesized that ectopic expression of a *dilp8* cDNA after the midthird instar transition checkpoint, a timepoint after which animals are no longer sensitive to the tissue damage-stress signal[34] (Fig. 1h), could rescue the increased AR phenotype of *dilp8* mutants (Fig. 3a). To control *dilp8* expression temporally, we placed a GAL4-dependent *dilp8* expression system (*tub > dilp8*) together with a ubiquitously-expressed temperature-sensitive GAL4-inhibitor, *tub-GAL80ts*, carried out a temperature switch after the midthird instar transition, and scored the timing of pupariation and puparium AR. As expected, the activation of *tub > dilp8* after the midthird instar transition did not delay the onset of metamorphosis (Fig. 3b), confirming that at this time-point Dilp8 is no longer able to signal via *R19B09 > -positive neurons to inhibit ecdysone biosynthesis and delay the onset of metamorphosis. However, activation of *tub > dilp8* after the midthird instar transition was sufficient to completely rescue the increased puparium AR of *dilp8* mutants (Fig. 3c). In contrast, activation of a mutant *dilp8* cDNA *dilp8^C150A*, which carries no Dilp8 activity due to the substitution of a critical cysteine to alanine[24], had no effect on puparium AR. These results are in line with the independence of the puparium AR phenotype on the *R19B09 > -positive neurons.

To genetically test for the spatial requirement of *dilp8* in the epidermis, we genetically knocked-down *dilp8* using the epidermal drivers *A58 >* and *Eip71CD > (A58 > dilp8-IR^TRIP* and *Eip71CD > dilp8-IR^TRIP*) and quantified puparium AR. However, neither condition altered the AR when compared to control genotypes (Fig. 3d, e). Attempts to use tissue-specific knockout of *dilp8* using a UAS-driven CRISPR-Cas9 system were unfortunately unsuccessful due to epistatic epidermal phenotypes caused by Cas9 expression (see Methods and Supplementary Fig. 3a, b). As puparium morphogenesis was particularly sensitive to *dilp8* levels, and incomplete loss or silencing of *dilp8* expression leads to normal puparium formation (Supplementary Fig. 1b-g), we hypothesized that in order to observe the *dilp8* knockout AR phenotype using the RNAi strategy, we would have to increase the strength of the RNAi in the epidermis. To do this, we combined the epidermal GAL4 drivers together (*A58 + Eip71CD > dilp8-IR^TRIP*). As expected, knockdown of *dilp8* using the combined drivers significantly increase puparium AR when compared to each control genotype (Fig. 3d, e). We conclude that epidermis-derived *dilp8* is required for proper puparium morphogenesis. Our results are strongly consistent with a model where the pupariation-associated upregulation of *dilp8* mRNA in the cuticle epidermis is the source of the Dilp8 peptide that signals via Lgr3 in *R18A01 > -positive neurons in the CNS.

*EcR* knockdown in the fat body using the *ppl > driver led to anterior retraction defects, which we hypothesized were due to

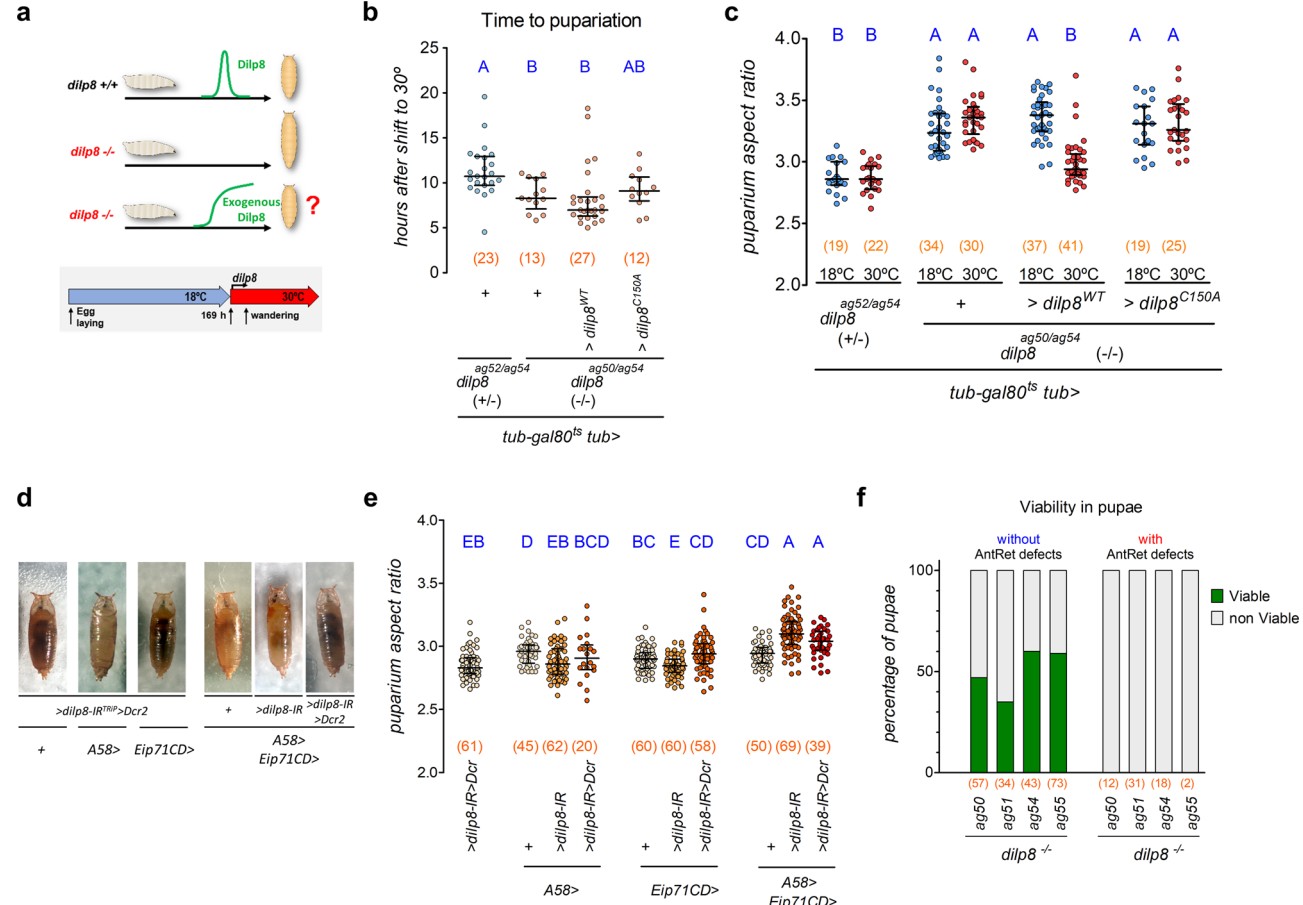

**Fig. 3 dilp8 is required in the cuticle epidermis during pupariation for puparium morphogenesis and viability. a** dilp8 temporal rescue scheme. **b** dilp8 expression after the midthird instar transition (tub > dilp8^WT at 30 °C) does not delay pupariation time. Shown are dot plots of time to pupariation. **c** dilp8 expression after the midthird instar transition rescues the puparium aspect ratio (AR) of dilp8 mutants. Dot plots showing puparium AR. **d** Representative photos of puparia from the depicted genotypes. **e** Knockdown of dilp8 using combined epidermal drivers increases the aspect ratio of puparia. The same batch of A58 > / + and Eip71CD > / + control animals were used for Fig. 2f. Dot plots showing puparium AR. **f** Percentage of viable pupae (green) with and without anterior retraction (AntRet) defects. Failure in AntRet decreases pupal viability. Statistics (full details in Supplementary Table 2): **b**, **c**, **e** Dots: one animal. Horizontal bar, median. Error bars: 25-75% percentiles. **b**, **c** Dunn's test. **e** Conover's test. **b**, **c**, **e** Same blue letter, P > 0.05. (N) Number of animals (orange).

effects of the EcR on other pathways (Fig. 2e, lower panels). To test for a role of dilp8 in the fat body or in any other ppl > -positive cell type, we knocked-down dilp8 using ppl > and quantified AR and looked for anterior retraction defects. ppl > dilp8-IR^TRIP did not increase puparium AR compared to controls, and had no detectable anterior retraction defects (Supplementary Fig. 3c, d). These results are consistent with our assumption that the anterior retraction defects caused by EcR knockdown in ppl > cells are not directly related to the Dilp8/Lgr3 pathway.

**Proper anterior retraction requires the Dilp8-Lgr3 pathway and is essential for survival.** While the puparium shape defect of dilp8 and Lgr3 mutants is evident at the population level, the phenotype follows a normal distribution and includes animals with puparium ARs overlapping the normal spectrum (e.g., see Fig. 1b, f). Likewise, failure to retract anterior segments is also incompletely penetrant, occurring in 5–40% animals, depending on the dilp8 allele (Supplementary Fig. 3e, f). dilp8 and Lgr3 mutants also show incomplete pupal viability (Supplementary Fig. 3g). Similar results were obtained by ubiquitous or pan-neuronal RNAi knockdown of Lgr3 (tub > Lgr3-IR or R57C10 > Lgr3-IR, respectively) confirming that the phenotype is specific for loss of Lgr3 activity in neurons (Supplementary Fig. 3h). To

test if this lethality was linked to puparium AR defects, we measured AR of puraria from animals that eclosed or not. Only one out of four dilp8 mutant genotypes surveyed showed a statistically significant difference between the puparium AR of animals that survived or died (Supplementary Fig. 3i). Hence, we conclude that there is no consistent association between survival and puparium AR. To test if this lethality was linked to anterior retraction defects, we followed pupal viability in animals with gross anterior retraction defects. We find that 100% of animals with visible anterior retraction defects fail to eclose, suggesting that proper anterior retraction is critical for pupal viability (Fig. 3f). Furthermore, ~50% of animals without clear anterior retraction defects also die. It is likely that those animals still have subtle anterior retraction defects (for example, they could be unable to seal the cuticle after retraction of the mouth hooks). Nevertheless, the fact that a fraction of mutants achieves WT-level puparium AR, at least something that looks like proper anterior retraction of the pre-spiracular segments, and survives, proves that the Dilp8-Lgr3 pathway is not per se the signal for anterior retraction. Rather, this suggests that the role played by Dilp8-Lgr3 in anterior retraction and proper puparium AR remodeling is modulatory and the mechanism might involve the setting of a threshold that defines a yes or no response (e.g.,

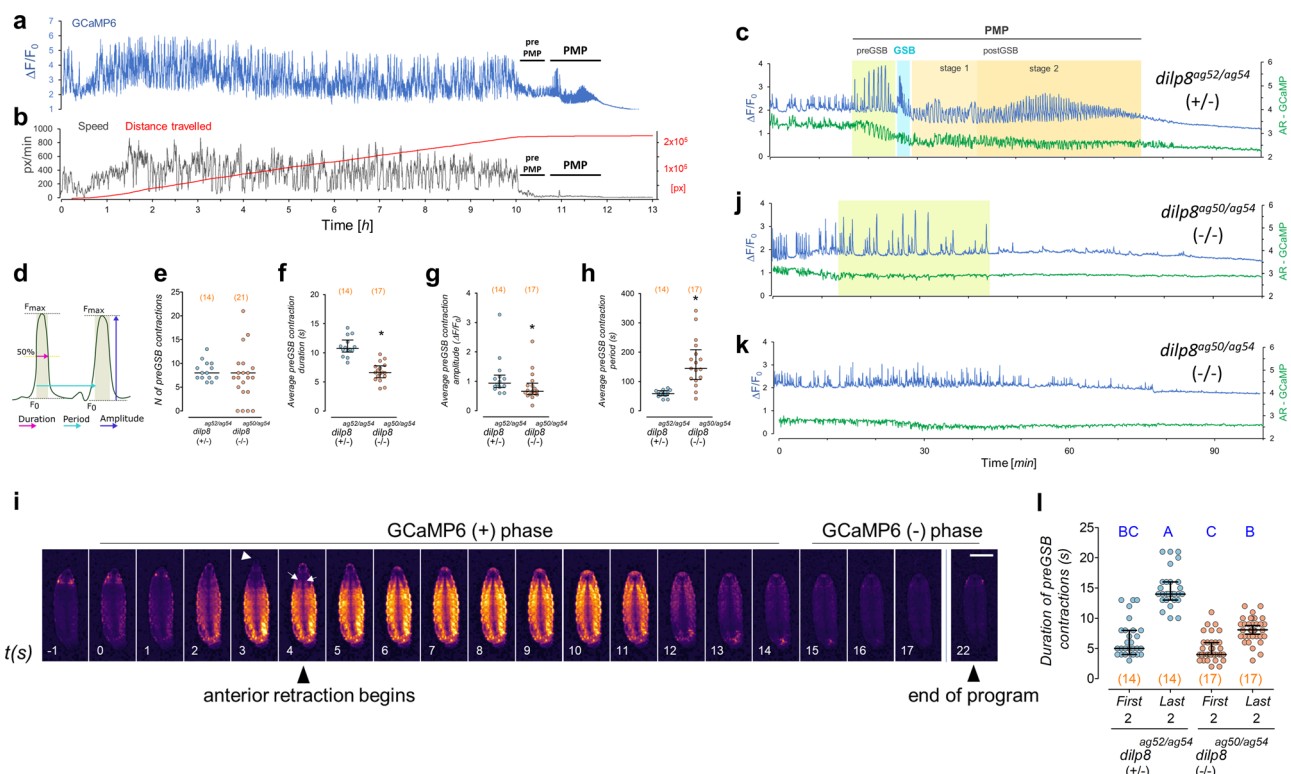

**Fig. 4 Dilp8 is critical for progression of the pupariation motor program. a** Muscle calcium (mhc»GCaMP) fluctuations of a single WT (*dilp8* +/−) larva (whole-body measurement, blue). Pupariation motor program (PMP). **b** Speed (black), and distance traveled by (red) the same larva depicted in **a**. **c** PMP in (**a**) and its specific stages. Shown are mhc»GCaMP (blue) and aspect ratio (AR-GCaMP, green) fluctuations. **d** Scheme describing the parameters measured for the pre-GSB contractions. **e** Dot plots showing the number of pre-GSB contractions of WT (*dilp8*+/−) and *dilp8* mutant (*dilp8*−/−) animals. **f–h** Dot plots showing the average **f** duration, **g** amplitude, and **h** period of pre-GSB contractions in WT and *dilp8* mutants. **i** Time-lapse of GCaMP oscillations during a WT pre-GSB contraction. Anteriormost segments are initially extruded (arrowhead) by the strong whole-body contraction and subsequently internalized by the activation of ventral longitudinal muscles (arrows). Representative profile from 3 recorded animals. **j** An example of muscle calcium (mhc»GCaMP) fluctuation (blue) and aspect ratio (AR-GCaMP, green) fluctuations of a *dilp8* mutant animal that showed pre-GSB-like contractions and one that **k** did not show any detectable pre-GSB contractions. **l** *dilp8* mutants fail to increase the duration of the pre-GSB contractions with time. Shown are dot plots of the duration of the first and last two pre-GSB contractions of WT and *dilp8* mutants. Statistics (full details in Supplementary Table 2): **e–h**, **l** Dot: average per larva. Horizontal bar, median. Error bars: 25–75%. **e**, **g**, **h** Mann–Whitney Rank sum test. **f** Student's *t*-test. **l** Dunn's test. Same blue letters, *P* > 0.05. *P < 0.05. P = 0.76 in **e** (excluding animals with no contractions). (N) Number of animals (orange). Scale bar, 1 mm.

proper anterior retraction or not) to an intensifying morphogenetic process. To learn more about the mechanism underlying the pupariation-specific defects of Dilp8-Lgr3 pathway mutants, we decided to observe pupariation directly.

**Direct observation of pupariation motor program (PMP) in pupariation arenas.** Whereas direct observation of pupariating animals under white light is informative, barometric measurement of internal pressure changes in pupariating *Sarcophaga bullata* animals has demonstrated complex pulsations that have been correlated with different muscle contraction programs[57]. In order to perform long-term live imaging and quantitative image analyses of the muscle contraction programs that characterize pupariating behavior, we constructed a series of raspberry-pi-based behavioral arenas (Supplementary Fig. 4a, see Methods) and monitored muscle contractions of animals using a GCaMP Calcium reporter [13XLexAop2-IVS-GCaMP6f-p10, ref. [58]] expressed under the control of a custom-engineered muscle-specific LexA driver, *mhc-LHV2*[59–62] (*mhc»GCaMP*, see Methods). *mhc»GCaMP* animals present bright muscle-contraction-dependent green fluorescence visible under blue light in dissecting scopes (Supplementary Fig. 4b; Supplementary Video 1). Monitoring of *mhc»GCaMP* animals in pupariation

arenas allowed precise quantitative assessment of *Drosophila* pupariation behavior (Fig. 4a, b; Supplementary Video 2). The first discernable feature of pupariation is the reduction in larval locomotion behavior that precedes the onset of the pupariation motor program (PMP) by ~53.9 (23.2–82.6) min or ~89.8 (59.3–130.6) min [median (25–75%)] depending on the genetic background (*dilp8*+/− or *Lgr*+/−, respectively) (Fig. 4a, b, Supplementary Fig. 4c, d). Monitoring of *dilp8* mutants carrying the *mhc»GCaMP* cassettes revealed no statistically significant difference in pre-PMP locomotor patterns (Supplementary Fig. 4c). Similar results were obtained for *Lgr3* mutants (Supplementary Fig. 4d). These results indicate that the pupariation problems that arise in animals lacking the Dilp8-Lgr3 pathway arise after the triggering of the pre-PMP and likely occur during the PMP itself.

The PMP comprises four distinguishable behavioral subunits (i.e., stereotyped motor programs): pre-GSB, GSB, post-GSB1, and post-GSB2, (Fig. 4c), which are labeled as a function of the most discernible pupariation subprogram: glue spreading behavior (GSB), a highly stereotyped short behavior where the animal spreads ventrally the secretory glue that is expelled from the salivary gland to promote adhesion of the puparium to the substrate. This and the other behavioral subunits of PMP are

further detailed in order of execution below, as well as how the absence of *dilp8* or *Lgr3* affects these subprograms.

The first stage of the PMP ("pre-GSB," Fig. 4c) is a 6.1 (5.8–9.4) min-long [median (25–75%)] motor program characterized by a series of increasingly strong whole-body contractions, which have a characteristic number, duration, amplitude, and period (Fig. 4d–h, Supplementary Fig. 4e–h and Supplementary Fig. 5). Each contraction starts with a quasi-synchronous body muscle contraction that vigorously reduces body AR, causing the transient extrusion of the anteriormost segments, which appear negative for mhc»GCaMP fluorescence at this stage, suggesting that the anterior segments are "squeezed" out by the increase in posterior pressure (Fig. 4i, Supplementary Video 3) After 2–4 s, large ventral intersegmental (longitudinal) muscles[63,64] are activated and gradually retract the anteriormost segments into the body, a process that helps extrude the anterior spiracles (Fig. 4i, Supplementary Video 3). Typically before the last major contraction, full and irreversible anterior retraction of the pre-spiracular segments is achieved, and the anterior segments are no longer extruded until the animal switches to the next behavioral subunit: that of glue expulsion and GSB, described in detail below.

**dilp8 and Lgr3 mutants have defective pre-GSB.** While all control animals surveyed performed pre-GSB, we failed to detect characteristic pre-GSB contractions in 4/21 (19.0%) *dilp8* mutants (Fig. 4e, j, k). Similar results were observed for *Lgr3* mutants (Supplementary Fig. 4e, j, k). The existence of a fraction of mutant animals that sclerotize their cuticle without any characteristic strong body remodeling contraction (Fig. 4k, Supplementary Fig. 4k), explains at least in part the very high puparium AR of some animals. The rest of the *dilp8* and *Lgr3* mutants showed pre-GSB-like mhc»GCaMP fluctuations that were indistinguishable in number from the pre-GSB contractions of their respective WT controls yet were significantly weaker, shorter, and more dispersed (Fig. 4e–h, Supplementary Fig. 4e–h, Supplementary Fig. 5).

**dilp8 and Lgr3 mutants do not perform GSB or post-GSB.** Whereas the pre-GSB phenotypes of *dilp8* and *Lgr3* mutants are incompletely penetrant, both mutants show a 100%-penetrant failure in progressing towards GSB, the next behavioral subunit of the PMP (Fig. 5a, Supplementary Video 3). The biosynthesis and secretion of secretory glue have been studied in the context of ecdysone signaling and protein trafficking, respectively[12–14,65]. However, little work has been done on the behavioral context of secretory glue expulsion per se since the description of the behavior and the function of the glue as a cementing agent by Gottfried Fraenkel and Victor Brookes in 1953[11]. Hence, the associated motor program of GSB has not been properly described. To describe GSB in further detail, we filmed the PMP of larvae expressing the salivary gland glue protein, Sgs3, translationally fused to GFP (Sgs3::GFP) under the control of its own regulatory regions[12]. GSB has two phases, an initial tetanic contraction phase that is followed by a series of peristaltic movements that promote the expulsion and the spreading of the secretory glue onto the ventral surface of the animal (Fig. 5b, Supplementary Videos 3, 5, 6). The specific and sustained contraction of ventral anterior segments ("ventral tetanus" in Fig. 5b), most noticeably A2, that initiates the GSB stage slightly arches the anterior half of the larva for 17–70 s, depending on the larva (Fig. 5b; Supplementary videos 5–7). This culminates with the initiation of an anterior peristaltic wave that propagates from T2 to A2 in ~3 s, further squeezing the anterior segments. This is followed closely (milliseconds) by the expulsion of the salivary gland contents (Fig. 5b). One or two seconds following glue expulsion, a series of coordinated peristaltic movements propagate forwards and backwards, starting from segment A2. These forth and back peristaltic movements slowly progress from A2 to posterior segments, reaching the final larval segments by the final waves (11–12 peristaltic waves in total) (Supplementary Videos 3, 5, 7, 8). Each wave contributes to spreading the glue towards the posterior ventral surface of the animal. During GSB, the animal typically moves forward ~half of its length, reaching its final pupariation site, where it typically waves its anterior end left and right a few times. This "head waving" marks the end of GSB. The total duration from the tetanus phase to the head waving is 71 s (62–86) or 63 s (56–68) [median (25–75%)], depending on the genetic background (*dilp8*(+/−) or *Lgr3* (+/−), respectively) (Fig. 5c).

To verify if GSB was a *D. melanogaster*-specific behavior, we monitored pupariating *Drosophila virilis* animals in our arena. *D. virilis* flies are predicted to have shared a last common ancestor with *D. melanogaster* about 50 MYA [confidence interval (38–62 MYA)][56]. Direct observation of GSB in *D. virilis* (Supplementary Video 9), suggests that the behavior has been conserved for at least 50 MY in *Drosophila*.

The next PMP behavioral subunit, named "post-GSB" typically lasts 51.3 min (45.3–60.47) or 46.4 min (41.5–50.0) [median (25–75%)] in total, depending on the genetic background (*dilp8* (+/−) or *Lgr3*(+/−), respectively), and is terminated by a gradual reduction in mhc»GCaMP-fluorescence fluctuations, which we can clearly associate with cuticle hardening, as the puparium AR no longer changes by the end of post-GSB (Figs. 4c and 5d, Supplementary Videos 7–8). *dilp8* and *Lgr3* mutants also show no visible signs of normal post-GSB (Fig. 4j, k, and 5e, Supplementary Fig. 4j, k). WT post-GSB can be divided into at least two stages that are characterized by different total mhc»GCaMP-fluorescence fluctuation patterns, post-GSB1 and post-GSB2. These stages divide post-GSB roughly in half. Both stages have complex contraction patterns, involving contraction of the whole body and the anterior longitudinal muscles. The first stage, post-GSB1, is characterized by longer, slightly stronger, and more separated contractions than the second stage, post-GSB2 (Supplementary Fig. 6a–f). The transition from one stage to the other is smooth, without a clear limit between them, and both types of contractions (post-GSB1 and 2) can coexist during the transition. Thus, we arbitrarily established a boundary between stages after the occurrence of typically one or two mhc»GCaMP-fluorescence peaks that were longer than preceding and subsequent ones. These contractions hence define the end of post-GSB1 and the beginning of post-GSB2 (Supplementary Fig. 6g, h).

Furthermore, we noticed that while the median duration of the pre-GSB contractions in WT animals increases ~10 s from the beginning to the end of the program (Fig. 4l, Supplementary Figs. 4l and 5), the duration of the pre-GSB contractions in mutant animals, despite also increasing with time, rarely achieved the values of control animals (Fig. 4l, Supplementary Figs. 4l and 5). Clearly, the pre-GSB program is abnormal in *dilp8* and *Lgr3* mutants. As the *dilp8* and *Lgr3* alleles assayed are genetic nulls, and a fraction of these nulls fails to perform pre-GSB, while the other fraction fails during it, these results are consistent with our hypothesis that the Dilp8-Lgr3 pathway regulates a thresholded morphogenetic mechanism slightly before or during pre-GSB. This suggests that the function of the Dilp8-Lgr3 pathway is to control the timing of when this threshold is reached during the PMP.

While post-GSB as a whole seems to contribute to the slight reduction in AR and maintenance of the remodeled puparium

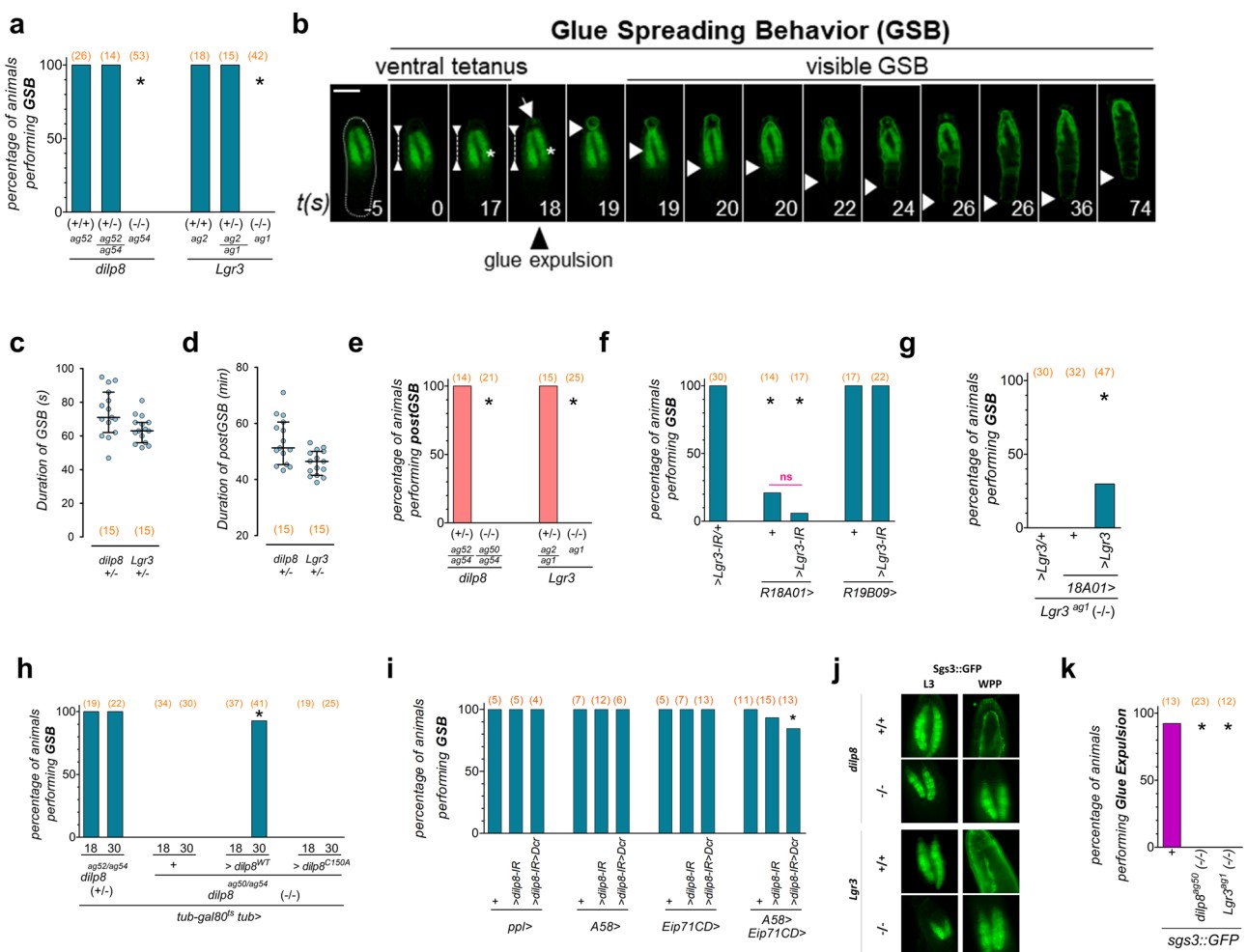

**Fig. 5 Dilp8-Lgr3 pathway is required for glue expulsion and spreading behavior. a** *dilp8* and *Lgr3* mutants do not perform GSB. Shown is the percentage of animals of the depicted genotypes that perform GSB. **b** Photo time-series of GSB and its two phases (ventral tetanus and visible GSB) in a larva expressing the salivary gland glue protein Sgs3::GFP (green) as a marker for glue (arrow, and descending white arrowhead marking progression of glue spreading towards the larval posterior, bottom). Representative images of 3 animals (see also Supplementary Videos 3, 5-7). **c** Dot blots showing the duration of GSB and **d** post-GSB in control animals of the depicted genotypes. **e** *dilp8* and *Lgr3* mutants do not perform post-GSB. Shown is the percentage of animals of the depicted genotypes that perform post-GSB. **f** Knockdown of *Lgr3* in *R18A01 >* neurons or *R18A01 >* alone, but not in *R19B09 >*, impedes GSB. Shown is the percentage of animals of the depicted genotypes that perform GSB. **g** Expression of *UAS-Lgr3* (*>Lgr3*) in *R18A01 >* neurons partially rescues the GSB defect of *Lgr3* mutants. Shown is the percentage of animals of the depicted genotypes that perform GSB. **h** GSB is rescued in *dilp8* mutants by expression of Dilp8 after the midthird instar transition. Shown is the percentage of animals of the depicted genotypes that perform GSB. **i** RNAi knockdown of *dilp8* using combined epidermal drivers (*A58 > +* *Eip71CD >*), but not each one alone, disrupts GSB in a fraction of animals. Shown is the percentage of animals of the depicted genotypes that perform GSB. **j** *dilp8* and *Lgr3* mutants fail to expulse glue (*Sgs3::GFP*, green). **k** Quantification of **j**. Shown is the percentage of animals of the depicted genotypes that perform glue expulsion. Statistics (full details in Supplementary Table 2): **a, e-i, k** Binomial tests with Bonferroni correction. **f** Fisher's Exact Test (magenta line). **c, d** Dots, one larvae. Horizontal bar, median. Error bars, 25-75%. *$P < 0.05$. ns, non-significant ($P > 0.05$). (N) Number of animals (orange). Scale bar, 1 mm.

shape, the functions of the post-GSB stages are not all clear. One important event that occurs during post-GSB is the formation of the operculum, from where the adult animal will exit the puparium when it is time to eclose. mhc»GCaMP monitoring shows that operculum formation is associated with strong tetanic contraction of at least three bilateral dorsoventral muscles in segments T2, T3, and A1 and at least two large dorsal longitudinal muscles, probably of segment A2 (Supplementary Fig. 6i, Supplementary Videos 7 and 8, min 5:40 and 4:45, respectively). Hence, operculum formation appears to be an active process requiring muscle activity, warranting further research on the regulation and evolution of this process. Clearly, this motor program is independent of *dilp8* or *Lgr3* and of progression to post-GSB, as it occurs normally in these mutants that do not perform the latter. The end of post-GSB and

operculum formation marks the end of the whole PMP program. At this stage, the animal has all characteristics of a WPP at T0, and the first signs of visible cuticle tanning are detectable within ~30 min.

**Lgr3 is required in R18A01 neurons for GSB.** To confirm that progression into GSB is also mediated by *R18A01 >*-positive neurons, we scored for the presence of GSB in *R18A01 > Lgr3-IR* animals. We find that *R18A01 > Lgr3-IR*, but not *R19B09 > Lgr3-IR*, completely abrogates GSB (Fig. 5f). However, *R18A01 >* alone also partially abrogates GSB. It is therefore possible that the *R18A01 >* insertion or the presence of an extra copy of the *R18A01* cis-regulatory-module itself interferes with GSB. As this is an *Lgr3* cis-regulatory-module, it could interfere with

endogenous *Lgr3* levels by acting as a sponge for rate-limiting transcription factors, for instance. If this were true, GSB should be rescuable in *Lgr3^ag1* animals using *R18A01 > Lgr3*, the same way that puparium AR was rescuable (Fig. 1l). However, we find that while *R18A01 > Lgr3* rescues AR, it only partially rescues GSB in *Lgr3^ag1* animals (Fig. 5g). The fact that the GSB rescue is incomplete could suggest that *Lgr3* is an exquisitely limiting factor in the presence of *R18A01>*. Alternatively, a second factor in the *R18A01 >* line could affect GSB but not AR, in an Lgr3-independent manner. For these reasons, conclusions on GSB based on the *R18A01 >* driver should be taken cautiously.

**Dilp8 is required in the epidermis for GSB**. To confirm that proper GSB requires the pupariation peak of *dilp8* in the epidermis, we carried out a temporal-rescue-experiment of *dilp8* mutant animals and a tissue-specific knockdown using RNAi and epidermal GAL4 drivers. We find that the temporally-controlled expression of *dilp8* after the midthird instar transition using the same *tub-Gal80ts, tub > dilp8* strategy that effectively rescues puparium AR of *dilp8* mutant animals (Fig. 2i), also rescued GSB in 38/41 animals (Fig. 5h). This result is consistent with the pupariation-associated Dilp8 peak being the source of the Dilp8 required for proper GSB.

Next, we knocked-down *dilp8* in the epidermis using the epidermal drivers *A58 >* and *Eip71CD > (A58 > dilp8-IR^TRIP* and *Eip71CD > dilp8-IR^TRIP*) or in the fat body using *ppl > (ppl > dilp8-IR^TRIP*) as a negative control, and scored for GSB. However, neither manipulation affected GSB (Fig. 5i). Hence, as we did for the AR experiments described above (Fig. 3e), we increased the GAL4 strength in the epidermis by combining both *A58 >* and *Eip71CD >* epidermal drivers with the *dilp8-IR^TRIP* transgene (*A58 + Eip71CD > dilp8-IR^TRIP*). In contrast to each GAL4 driver alone, this manipulation abrogated GSB in 6.7% (1/15) and 15.4% (2/13) of animals in the absence or presence of the *UAS-Dcr* cassette, respectively, whereas 0/75 animals of 10 control genotypes failed in GSB (Fig. 5i). We conclude that *dilp8* is required in the epidermis for GSB and that very few *dilp8* molecules must be sufficient for proper pupariation progression.

As the genetic knockdown of EcR in the epidermis (*A58 > EcR-IR* or *Eip71CD > EcR-IR)* significantly reduced *dilp8* mRNA levels, we also assayed for GSB in these animals. However, knockdown of EcR in the epidermis did not interfere with GSB (Supplementary Fig. 7a). This is consistent with our findings that neither genotype completely eliminated *dilp8* transcript levels (Fig. 2g), and is in line with the model where the epidermally-derived Dilp8 is required downstream of ecdysone-signaling for proper GSB.

**The Dilp8-Lgr3 pathway is required for glue expulsion**. As glue expulsion and GSB are intimately linked, and both *dilp8* and *Lgr3* mutants completely fail in performing the latter, we verified if glue expulsion was also affected by monitoring Sgs3::GFP localization in each mutant before and after pupariation (L3 wandering stage and WPP T0). Results showed that Sgs3::GFP is expulsed onto the ventral side of control WPP T0 animals, as expected, but is retained in the salivary glands of *dilp8* and *Lgr3* mutants at WPP T0 (Fig. 5j, k). Close inspection of dissected salivary glands showed that Sgs3::GFP is properly secreted into the lumen of the glands in *dilp8* and *Lgr3* WPP T0 mutants (Supplementary Fig. 7b), showing that the initial steps of glue production and secretion are unaffected in *dilp8* and *Lgr3* mutants. These results demonstrate that the Dilp8-Lgr3 pathway is required for glue expulsion and GSB.

**GSB occurs independently of glue expulsion**. The fact that glue expulsion fails in *dilp8* and *Lgr3* mutants could have implications

for the observed pupariation phenotypes. For instance, the persistence of the enlarged salivary glands in the body could hinder body contractions, leading to increased AR. Also, the fact that glue expulsion precedes most of the stereotypic peristaltic movements of GSB, could mean that both processes are mechanistically linked. For instance, GSB could require previous glue expulsion, i.e., GSB could be a response to either external sensing of the expelled glue, or of a strong reduction in internal body pressure linked with the expulsion of the copious amounts of secretory glue. Alternatively, glue expulsion could occur independently of GSB or even be a consequence of the GSB program. To gain insight into this relationship, we hypothesized that glue expulsion was required for GSB. To test this, we performed RNAi-knockdown of the Rho GTPase Rho1 using the salivary-gland specific driver *forkhead-GAL4 (fkh>)*. This genetic manipulation has been shown to completely block glue secretion to the lumen of the salivary gland, and hence eliminate glue expulsion[65]. We thus expected that *fkh > Rho1-IR* animals would not perform GSB. To control for the efficiency of the *fkh > Rho1-IR* genetic manipulation we monitored glue-expulsion dynamics using the Sgs3::GFP reporter. As expected, *fkh > Rho1-IR* animals completely failed in glue expulsion, retaining Sgs3::GFP in their salivary glands past the pupariation phase (Supplementary Fig. 7c). However, in contrast to our hypothesis, *fkh > Rho1-IR* animals executed GSB just as control *fkh >* animals did (Supplementary Fig. 7d) and even generated puparia with normal ARs (Supplementary Fig. 7e). These results demonstrated that retention of the enlarged salivary glands does not interfere with the PMP or puparium morphogenesis. We conclude that GSB occurs independently of glue expulsion. A likely scenario is that glue expulsion is triggered by the first peristaltic wave of GSB following the tetanus phase (Fig. 5b). In this case, GSB should be best defined as "glue expulsion and spreading behavior."

**The Dilp8-Lgr3 pathway antagonizes cuticle sclerotization during the PMP**. One possibility that arises from the experiments described above is that *dilp8* mutants fail to progress in their PMP due to increased resistance of the cuticle to muscle contraction. To gain further insight into this possibility, we calculated the total PMP based on two parameters: the total duration of detectable mhc»GCaMP fluctuations from the initiation of PMP (pre-GSB) up to the cessation/stabilization of mhc»GCaMP fluctuations, and the time it takes for the animal to cease actual AR-affecting body contractions (i.e., the time from pre-GSB to complete body immobilization/sclerotization). Strikingly, whereas there was no difference in the total PMP time between *dilp8* mutants and controls as evaluated by mhc»GCaMP fluctuations, the puparium AR of *dilp8* mutants stabilized ~25 min earlier than that of controls (Fig. 6a). These results strongly suggest that the cuticle of *dilp8* mutants is hardening precociously.

If the function of the Dilp8-Lgr3 pathway is to transiently postpone cuticle sclerotization during the initial stages of the PMP, then it follows that excessive Dilp8 signaling could lead to a delay in cuticle sclerotization. To test this, we quantified the duration from GSB to detectable cuticle tanning (used here as a surrogate for sclerotization) in the *dilp8* mutants that were rescued at wandering stage with *tub > dilp8* (Fig. 5h). Results showed that *tub > dilp8*-rescued *dilp8* mutants took 31 min longer to tan than control WT animals (Fig. 6b). Tanning was delayed by >100 min in some animals (Fig. 6b). A fraction of rescued animals even executed parts of the PMP twice in tandem (Supplementary Video 10). These animals expressed crawling behavior at a time when the cuticle should have been sclerotized. Importantly, removal of animals with double GSBs or of all the animals with extreme PMP-duration values from analyses still revealed

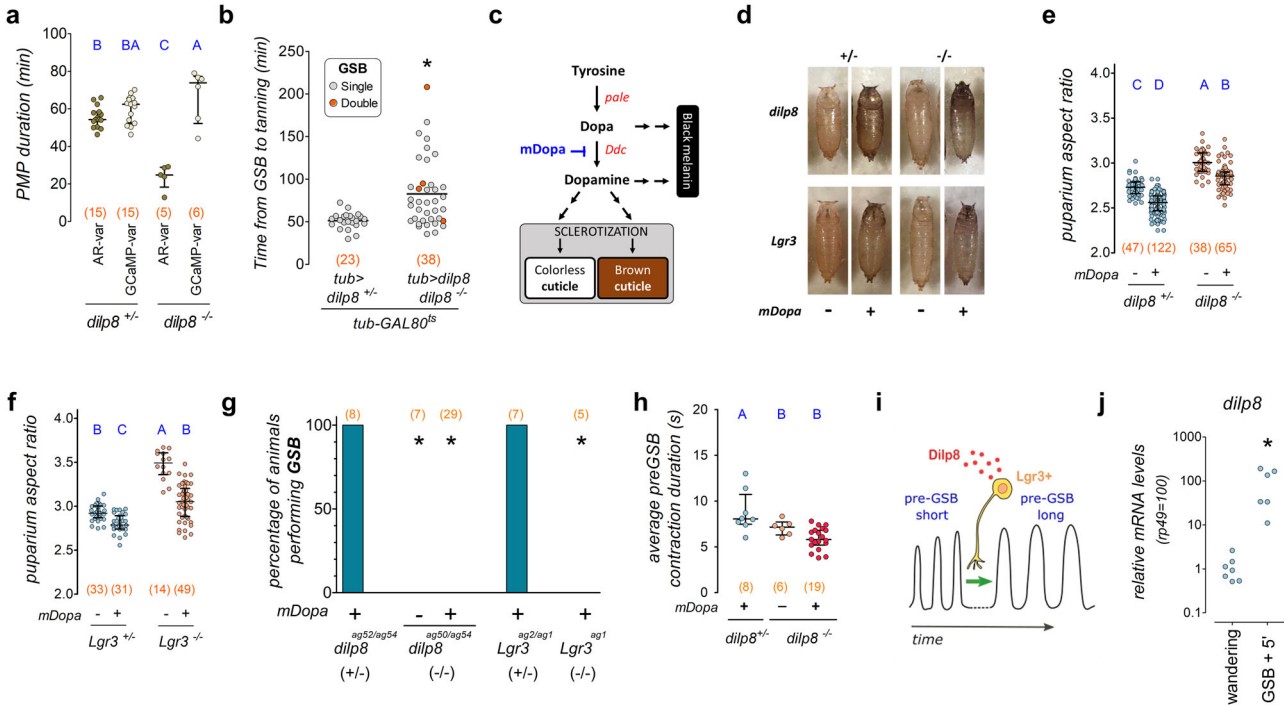

**Fig. 6 Pupariation progression by coupling morphogenetic and neuromotor subprograms. a** *dilp8*-mutant puparium aspect ratio (AR) fluctuations are briefer than muscle calcium (mhc»GCaMP) fluctuations during pupariation. Shown are dot plots of PMP duration in *dilp8* mutants (−/−) and controls (+/−) according to variation in puparium AR (AR-var) or mhc»GCaMP (GCaMP-var). *dilp8*(−/−) is *dilp8*[ag52/ag54]. *dilp8*(+/−) is *dilp8*[ag52/ag54]. **b** Post-midthird instar transition-expression of *tub > dilp8* delays tanning. Shown are dot plots of the time from GSB to tanning. Red dots, animals performing two GSBs. **c** Cuticle sclerotization and tanning pathway. mDopa, α-methyldopa. **d** Photos of puparia. Effects of α-methyldopa. **e** Quantification of **d** for *dilp8* and **f** *Lgr3* mutants and controls. Shown are dot plots of puparium AR. **g** α-methyldopa treatment does not rescue GSB of *dilp8* or *Lgr3* mutants. Shown is the percentage of animals of the depicted genotypes that perform GSB. **h** α-methyldopa treatment does not rescue the average duration of pre-GSB contractions of *dilp8* mutants. Shown are dot plots of the average pre-GSB contraction duration. **i** Model for the Dilp8-Lgr3-dependent modulation of pre-GSB. **j** *dilp8* mRNA levels increase 5 min after GSB. Shown are qRT-PCR estimations of *dilp8* mRNA levels in WT animals. Statistics (full details in Supplementary Table 2): **a**, **b**, **e**, **f**, **j** Dots: one animal. **h** Dots: average per animal. **a**, **e**, **f**, **h** Horizontal bar, median. Error bars, 25-75%. **a**, **e**, **h** Student–Neuwan–Keuls test. **f** Dunn's test. **b**, **j** Mann–Whitney Rank sum test. **g** Binomial tests with Bonferroni correction. **a**, **e**, **f-h** Same blue letters, *P* > 0.05. *P* < 0.05. (N) Number of animals (orange).

significantly-prolonged PMPs caused by *tub > dilp8* activation in wandering stage animals (28 or 12 min longer, respectively; Supplementary Fig. 8a, b). These results demonstrate that the PMP and cuticle sclerotization have been uncoupled by ectopic Dilp8 signaling and are consistent with the results indicating precocious sclerotization in *dilp8* and *Lgr3* mutants.

To independently confirm that the function of the Dilp8-Lgr3 pathway during pupariation is to transiently postpone cuticle sclerotization during the initial stages of PMP, we hypothesized that suppression of cuticle sclerotization would rescue all pupariation-related phenotypes of *dilp8* mutants. To do this, we fed α-methyldopa to *dilp8*- or *Lgr3*-mutant third-instar larvae in a concentration that attenuates cuticle sclerotization[66]. α-Methyldopa inhibits the enzyme Dopa decarboxylase (Ddc), which converts DOPA to dopamine in the epidermis, an essential step in insect cuticle sclerotization[67,68] (Fig. 6c). α-Methyldopa treatment is thus expected to have at least two effects: to inhibit cuticle sclerotization by reducing the amount of available Dopamine that gets fed into the cuticle sclerotization pathways, and a strong melanization of the cuticle, as the unconverted excess of the Dopamine precursor, DOPA, becomes available to the alternative black-melanin production pathway (Fig. 6c). Cuticle melanization per se is not expected to interfere with pupariation. As expected, α-methyldopa treatment led to strong melanization of the cuticle, confirming that Ddc was efficiently inhibited (Fig. 6c, d). As predicted, α-methyldopa treatment reduced puparium AR in *dilp8*

(Fig. 6e) and *Lgr3* mutants (Fig. 6f). Puparium AR was also reduced, albeit to a lesser extent, in the background controls of both mutants (Fig. 6e, f). Hence, one of the reasons why *dilp8* and *Lgr3* mutants do not achieve proper puparium AR is an excess of dopamine-mediated cuticle sclerotization, which increases the resistance of the cuticle to underlying muscle contractions. These results also suggest that in WT animals, cuticle sclerotization must start before the PMP as it contributes as a resistance force to the body-reshaping muscle contractions of the PMP. However, α-methyldopa-fed mutants still had anterior-retraction defects and did not achieve the same AR as controls (Fig. 6d–f), suggesting that rescue by α-methyldopa treatment was not complete. These findings indicate that *dilp8* and *Lgr3* played additional roles during pupariation.

**The Dilp8-Lgr3 pathway modulates the pre-GSB motor program.** To gain insight into this second mechanism, we monitored mhc»GCaMP6 in α-methyldopa-fed and vehicle-fed control animals. While α-methyldopa-fed WT animals performed all stages of PMP, including GSB, similarly to control animals (Fig. 6g, Supplementary Fig. 8c), α-methyldopa-fed *dilp8* and *Lgr3* mutants did not, remaining instead trapped in a pre-GSB-like phase, never switching to GSB (Fig. 6g, Supplementary Fig. 8d, e). α-Methyldopa treatment strongly increased the number of detectable pre-GSB contractions (Supplementary Fig. 8f) and mildly reduced their period (Supplementary Fig. 8g). This

demonstrates that cuticle sclerotization negatively affects puparium AR by antagonizing pre-GSB number and frequency. The critical finding regarding the second mechanism, however, was that α-methyldopa treatment had little or no effect on pre-GSB contraction duration relative to untreated *dilp8* mutants (Fig. 6h, Supplementary Fig. 8d, e), which should increase ~10 s toward the end of the pre-GSB phase, as it does in WT animals, before anterior retraction and GSB (Fig. 4l, Supplementary Fig. 4l). This leads to a model where *dilp8* mutants are locked in an early, *dilp8*-independent pre-GSB-like state, which we named pre-GSB$^{short}$. Dilp8-Lgr3 signaling is thus required to convert the pre-GSB$^{short}$ into the longer and stronger pre-GSB contractions, which we named pre-GSB$^{long}$, that typically occur at the end of the pre-GSB stage and that do not occur in *dilp8* or *Lgr3* mutants (Fig. 6i). Hence, we propose that successful anterior retraction requires both a Dilp8-dependent transient inhibition of cuticle sclerotization and the neuromodulation of the pre-GSB neuromotor contraction circuit from pre-GSB$^{short}$ to pre-GSB$^{long}$. While pre-GSB$^{short}$ can achieve some remodeling of the body it is ineffective in achieving successful anterior retraction and promoting the transit into the glue expulsion and spreading behavior phase. We further propose that successful anterior retraction is a gate to unlock the next behavioral subunit, GSB.

In order to transiently inhibit cuticle sclerotization and modulate the pre-GSB motor program, so that an effective anterior retraction is achieved, some Dilp8 protein would have to be present before the initiation of the pre-GSB program. We have shown that the peak in *dilp8* transcripts occurs around T0 (Fig. 2a), which occurs ~45–60 min after the initiation of pre-GSB (Fig. 4c). As we know that at −4 h before T0, *dilp8* mRNA levels are still flat (Supplementary Fig. 2a, c)[54], the 20HE-dependent *dilp8* upregulation must start between −4 h and T0, which is confirmed by the strong upregulation found in pre-WPP animals (Fig. 2a). However, pre-WPP can be anywhere between this ~1-h interval. To test if the *dilp8* transcripts are upregulated before T0 in a more precise manner, we obtained samples from whole animals exactly 5 min after they had performed GSB, a behavior that can be unequivocally scored, and compared *dilp8* mRNA levels to wandering L3 larvae by qRT-PCR. Results showed that *dilp8* mRNA levels were already upregulated by >2 orders of magnitude 5 min after GSB (Fig. 6j). This is consistent with the idea that enough Dilp8 protein is available for signaling events occurring 10–15 min before this time point, which corresponds to the onset of pre-GSB. This is also in line with our observations in *C. capitata*, where *cilp8* mRNA levels are already increased by a factor of ~88 in animals, which could be unequivocally-collected by eye at the ~5-min-long "body contraction" stage (Fig. 2i). Due to the obvious similarities, we assume that the *C. capitata* contraction phase corresponds to the pre-GSB stage of *D. melanogaster*. We conclude that the timing of the *ilp8* transcriptional peak is consistent with its proposed early time-window of activity during pupariation to promote PMP progression.

**Lgr3 is required in 6 ventral nerve cord neurons for PMP progression**. To try to further pinpoint which subpopulation of neurons is critical for proper pupariation, we took advantage of a serendipitous finding: while screening GAL4 lines for another *Lgr3*-dependent phenotype (coupling of growth and maturation), we observed elongated puparia when removing *Lgr3* using the line *R48H10-GAL4* (*R48H10 >* )[51] (Fig. 7a). *R48H10 > Lgr3-IR* also disrupted GSB in 100% of the animals (Fig. 7b), suggesting that *R48H10 >* was active in the same cells as *R18A01 >* . The relatively sparse expression pattern of the *R48H10 >* driver[51], makes it valuable for intersectional genetics. In fact, only six *R48H10 >* -positive cells in the thoracic region of the CNS

expressed detectable levels of Lgr3 protein, as measured by an endogenously labeled Lgr3 translational reporter [*sfGFP::Lgr3$^{ag5}$*, ref. [26]] (Fig. 7c). Interestingly, six similar cells were amongst the co-labeled cells when the *R18A01 >* line was crossed with *UAS-CD8::RFP* and *sfGFP::Lgr3$^{ag5}$* (Supplementary Fig. 9a).

To genetically confirm that these 6 neurons were co-labeled by both *R18A01 >* and *R48H10 >* , we generated a genetic intersection between *R18A01* and *R48H10* using a flip-out recombinase method[69] and an *R18A01-LexA* line (*R18A01»*, see Methods, Supplementary Fig. 9b–d). This intersection, hereafter described as *R18A01 ∩ R48H10*, allowed versatile usage of different UAS transgenes. As predicted from the patterns described above, *R18A01 ∩ R48H10 > CD8:GFP* consistently labeled the expected 6 VNC neurons (Fig. 7d–f). The soma of the 3 paired VNC neurons are located towards the midline of the boundaries of the T1p/T2a, and in the T2p, and T3p segments, respectively. The most anterior pair of labeled neurons has been previously described as the Midline Internal Lgr3-positive (MIL) neurons[26]. The two other pairs have not, to the best of our knowledge, been described in detail, but are always positioned ventrally, so we called them Ventral Midline Lgr3-positive (VML) neurons.

To confirm that these 6 VNC neurons require Lgr3 to promote PMP progression we used the *R18A01 ∩ R48H10 >* intersectional driver to drive *Lgr3* RNAi, and scored for puparium AR and the presence of GSB. Results revealed that *R18A01 ∩ R48H10 > Lgr3-IR* animals had increased puparium AR when compared to controls (Fig. 7g, h), and did not perform GSB (Fig. 7i), consistent with the requirement of the 6 VNC neurons for Dilp8 signaling. While all controls behaved as expected for puparium AR, the LexA version of the R18A01 driver, *R18A01»*, alone interfered with GSB (Fig. 7i). This interference was even stronger than the one found using the GAL4 version of this driver, *R18A01 >* (Fig. 5f), which is inserted in a different genome location (*attp2*), excluding insertional artefacts as a cause of the GSB interference. This confirmed our suspicion that the extra copy of the *R18A01* cis-regulatory-module per se interferes with normal pupariation. We again attempted to rescue the AR and GSB of *Lgr3$^{ag1}$* mutants by expressing *UAS-Lgr3* under the control of *R18A01 ∩ R48H10 >* . Results showed that *R18A01 ∩ R48H10 > Lgr3* rescued puparium AR, but not GSB (Fig. 7j, k). Hence, *R18A01»* is epistatic to *Lgr3* in GSB. To exclude the unlikely possibility that GSB is independent of the status of *Lgr3* in the 6VNC neurons, we attempted to rescue puparium AR and GSB in *Lgr3$^{ag1}$* mutants using *R48H10 > Lgr3* alone. The results of this rescue experiment clearly show that *R48H10 > Lgr3* fully rescues puparium AR and GSB in *Lgr3$^{ag1}$* mutants (Fig. 7l, m). Hence, we conclude that the *R18A01* cis-regulatory-module interferes with GSB specifically and epistatically to *Lgr3* function. Furthermore, we conclude that the six *R18A01 ∩ R48H10 >* -positive VNC neurons or a subset of them are the critical cells requiring Lgr3 to transduce the cuticle epidermis-derived Dilp8 signal at pupariation to promote PMP progression from pre-GSB into GSB.

**Lgr3 activity in pupariation-controlling neurons do not affect ecdysone biosynthesis or activity**. Above, we provide evidence that 20HE acts directly on the epidermis to induce *dilp8* transcription, placing *dilp8* downstream of 20HE signaling. Interestingly, this is conceptually the opposite of what Dilp8 does prior to the midthird instar transition checkpoint, where it acts upstream of 20HE production, inhibiting it[23–28,34,46]. However, it remained possible that Dilp8 also acts upstream of 20HE during pupariation, if the function of Dilp8-Lgr3 were to be, for instance, to inhibit vestigial 20HE signaling, contributing to the termination of the 20HE peak. To address this directly, we performed qRT-PCR of mRNA isolated from synchronized WPP T0 animals

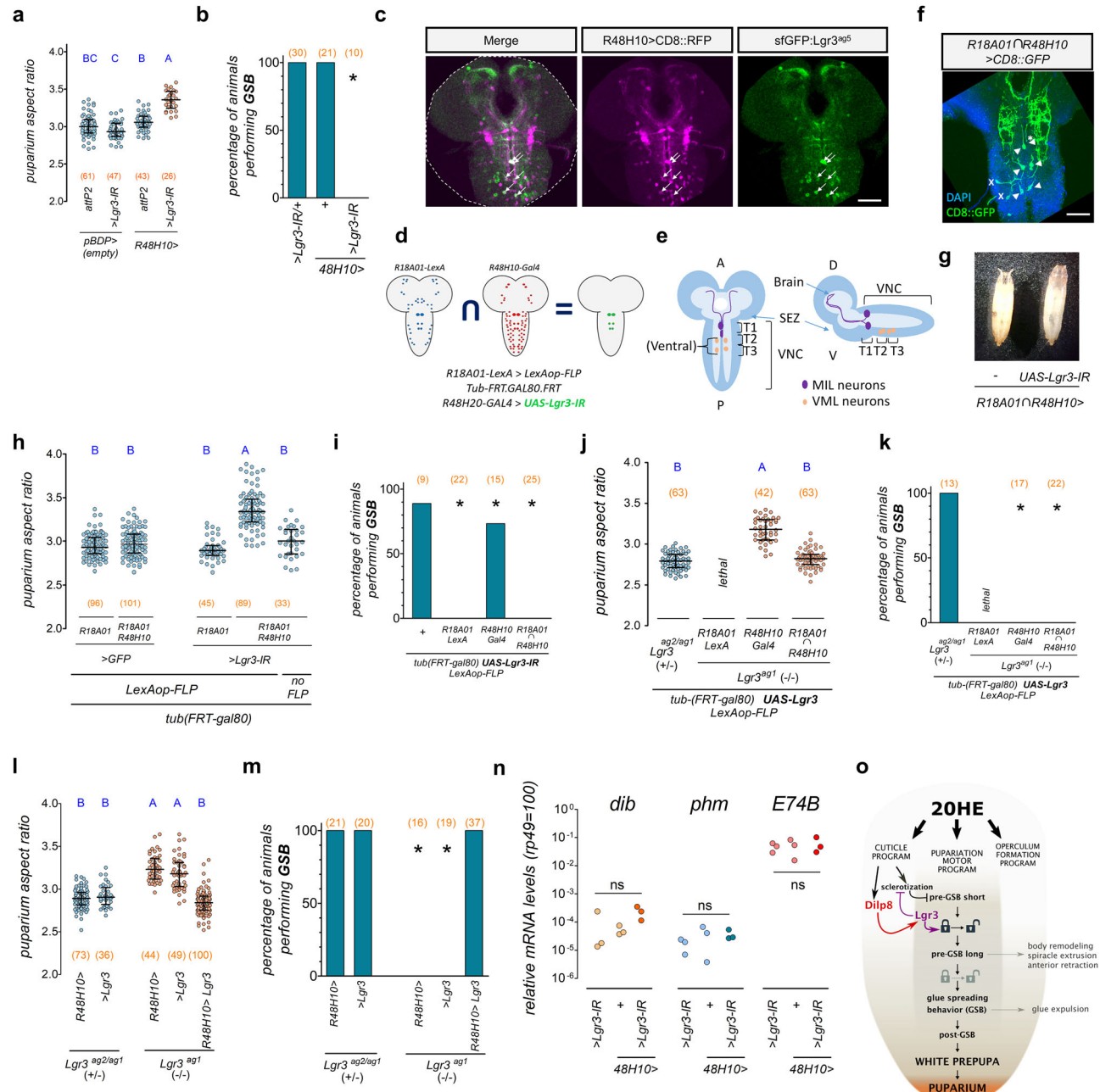

**Fig. 7 Lgr3 is required in six thoracic interneurons for PMP progression. a** *Lgr3* knockdown in *R48H10 >* neurons increases puparium aspect ratio (AR). Shown are dot plots of puparium AR. **b** *Lgr3* knockdown in *R48H10 >* neurons impedes GSB. Shown is the percentage of animals of the depicted genotypes that perform GSB. >*Lgr3-IR/+* data, same as Fig. 5f. **c** Six thoracic (6VNC) interneurons (white arrows) co-express *R48H10 > CD8::RFP* (magenta) and *sfGFP::Lgr3ag5* (anti-GFP, green). **d** *R18A01 ∩ R48H10* intersectional genetics system. **e** Cartoon of the 6VNC *R18A01 ∩ R48H10* neurons. SEZ, subesophageal zone. T1-3, thoracic segments. **f** 6VNC neurons (arrowheads) express *R18A01 ∩ R48H10 > CD8::GFP* (green). DAPI, blue. Asterisk, non-visible MIL neuron. X, non-reproducible cells. **g** Photos of control and *R18A01 ∩ R48H10 > Lgr3-IR* puparia. **h** *R18A01 ∩ R48H10 > Lgr3-IR* increases puparium AR. Shown are dot plots of puparium AR. Quantification of (**g**). **i** *R18A01 ∩ R48H10 > Lgr3-IR* abrogates GSB. *R18A01-LexA* (*R18A01»*) alone abrogates GSB. Shown is the percentage of animals of the depicted genotypes that perform GSB. **j** *Lgr3* expression (*UAS-Lgr3*) in *R18A01 ∩ R48H10* neurons rescues puparium AR in *Lgr3* mutants. Shown are dot plots of puparium AR. **k** *Lgr3* expression in *R18A01 ∩ R48H10* neurons does not rescue GSB. Shown is the percentage of animals of the depicted genotypes that perform GSB. **l** *Lgr3* expression in *R48H10* neurons rescues puparium AR in *Lgr3* mutants. Shown are dot plots of puparium AR. **m** *Lgr3* expression in *R48H10* neurons rescues GSB in *Lgr3* mutants. Shown is the percentage of animals of the depicted genotypes that perform GSB. **n** *R48H10 > Lgr3-IR* does not alter *phm*, *dib*, or *E74B* mRNA levels in WPP T0 animals. Dot plots showing qRT-PCR estimations of the depicted genes. **o** Model: Dilp8-Lgr3 pathway promotes pupariation program progression. Statistics (full details in Supplementary Table 2): **a**, **h**, **j**, **l** Dots: one animal. Horizontal bar, median. Error bars, 25-75%. **n** Dots: biological repeats. **a**, **h**, **j**, **l**, **n** ANOVA, followed by **a** Holm-Sidak's test. **h**, **j**, **l** Dunn's test. **n** ns not-significant. **b**, **l**, **k**, **m** Binomial tests with Bonferroni correction. Same blue letters, corrected *P > 0.05*. (N) Number of animals (orange). Scale bars, 50 µm.

lacking *Lgr3* activity in the R48H10 neurons (*R48H10 > Lgr3-IR*), which have aberrant puparium AR and do not perform GSB (Fig. 7a, b), and measured the relative mRNA levels of the ecdysone biosynthesis genes *phantom* (*phm*)[70] and *disembodied* (*dib*)[71] and the *EcR*-responsive gene, *E74B*[72]. Importantly, the *R48H10 > Lgr3-IR* condition was specifically chosen to avoid epistatic effects of the *R18A01 >* genotype or confounding factors that could be associated with the altered timing of the onset of pupariation when using *dilp8* or *Lgr3* mutations [a 3–4-h anticipation of pupariation occurs in the latter genotypes[23,26], which is attributable to pre-midthird instar transition effects of those genes, as this anticipation is not rescued by post-midthird instar transition expression of *tub > dilp8* (Fig. 3b)]. As expected, the qRT-PCR results showed no statistically significant difference in the transcript levels of *phm*, *dib*, or *E74B* between animals with *R48H10 > Lgr3-IR* and controls (Fig. 7n). These results suggest that there is no overt alteration of ecdysone signaling per se when the PMP-promoting Dilp8-Lgr3 pathway is abrogated. Hence, we conclude that the Dilp8-Lgr3 pathway acts downstream of 20HE to control the puparium motor program.

## Discussion

Here, we have found that the relaxin-like Dilp8-Lgr3 pathway, which has been previously shown to coordinate growth and maturation timing in earlier stages of third instar larvae[23–28,34,46], acts in a spatially- and temporally-independent manner during pupariation to promote pupariation motor program (PMP) progression. Epidermis-to-interneuron Dilp8-Lgr3 signaling couples peripheral tissue morphogenesis with centrally-controlled motor programs to promote progression from pre-"glue (expulsion) and spreading behavior" (pre-GSB) to "glue (expulsion) and spreading behavior" (GSB), which are the first and second behavioral subunits of the PMP. This is achieved by at least two parallel activities: by the transient inhibition of cuticle sclerotization, which promotes cuticle malleability, decreasing the resistance of the cuticle to the underlying muscle contractions, and by the neuromodulation of the Dilp8-independent pre-GSB$^{short}$ program to a Dilp8-dependent anterior-retraction-promoting pre-GSB$^{long}$ program. We hypothesize that both of these activities are necessary for the animal to transit from pre-GSB to the GSB phase (Fig. 7o).

We show that during pupariation, *dilp8* transcription is triggered as a response to ecdysone signaling in the cuticle epidermis. A similar conclusion was reached in a recent study focusing on the role of *dilp8* on terminal imaginal disc growth regulation[73], wherein *dilp8* was placed downstream of EcR in the cuticle epidermis during pupariation, strongly supporting our findings. When imaginal discs are abnormally growing in 3$^{rd}$ instar larvae, the Dilp8-Lgr3 pathway acts by antagonizing ecdysone biosynthesis, delaying the onset of pupariation[23–28,34,46]. Here, by knocking-down *Lgr3* activity in the critical 6VNC neurons that affect pupariation motor program progression, we find no evidence for altered levels of ecdysone biosynthesis or activity at the time when the Dilp8 peak is maximal, WPP T0. These results favor a model where the Dilp8-Lgr3 pathway acts downstream of 20HE signaling, which is conceptually the opposite of what Dilp8 does prior to the midthird instar transition checkpoint, where it acts upstream of 20HE production, inhibiting it[23–28,34,46]. It is also important to consider that Dilp8-Lgr3 signaling during pupariation controls at least two biological processes: cuticle sclerotization timing and pre-GSB neuromodulation. While both processes can be controlled by the 6 Lgr3-positive VNC neurons or by subsets of them, it is also possible that Dilp8-Lgr3 controls a third uncharacterized factor that acts upstream of these processes.

Several decades ago, the insect physiologist Gottfried Fraenkel and colleagues described the "pupariation factors"[7,74]. These are

factors of peptidic nature that controlled different subprograms of pupariation downstream of the steroid hormone ecdysone in the gray flesh fly, *Sarchophaga bullata*. A pyrokinin peptide has been biochemically identified as a factor capable of accelerating pupariation initiation[22], however, its requirement in vivo remains to be genetically demonstrated. The identification of Dilp8 as a pupariation factor with a genetically defined temporal and spatial role in *Drosophila* might pave the way for further identification of pupariation factors. It is unclear if Dilp8 corresponds to any of the proposed pupariation factors by Fraenkel, but it is not so dissimilar from PIF (puparium immobilization factor), due to similar profiles of expression[75]. This is further substantiated by the fact that PIF was proposed to be identical to ARF (anterior retraction factor) (a neurotropic factor that "releases behavioral patterns initiating pupariation, namely retraction of the three anterior segments bearing the cephalopharyngeal apparatus"[75], and that we show that the neurotropic peptide Dilp8 is required for fruitful anterior retraction in our study. This hypothesis is compatible with the fact that the order of body contraction and anterior retraction is inversed in *S. bullata* respective to *Drosophila*, yet the pupariation factors PIF/ARF act in a species-unspecific manner. Hence, PIF/ARF might indeed release anterior retraction after body contraction in *Drosophila*, which can be what Dilp8 does by promoting transition from pre-GSB$^{short}$ to pre-GSB$^{long}$. Hence, Dilp8 might as well be PIF/ARF. We hope that our work will stimulate further evo-devo studies and allow the molecular and genetic characterization of Fraenkel's pupariation factors.

Our work, together with previous work on the role of the Dilp8-Lgr3 pathway in growth and developmental timing coordination[23–28,34,46], suggests that this *Drosophila* relaxin pathway can be interpreted as a bona-fide heterochronic pathway, i.e., a pathway that controls the timing and/or duration of developmental processes. Heterochronic pathway genes are thought to partially contribute to the timely coordination of such programs by determining the timing of cell fate decisions cell-autonomously[76,77]. In addition, certain hormones, such as dafachronic acid, ecdysone, and thyroid hormones, also show heterochronic-like activities by orchestrating the timing of major life history transitions, non-cell-autonomously[76,78–80]. Not surprisingly, these activities were revealed in animals with major clear-cut transitions, such as those undergoing metamorphosis (e.g., flies and frogs). It is unclear how other animals achieve time coordination, especially when this coordination is restricted to a subset of organs in the body. Interestingly, relaxin and relaxin-like signaling have been linked to complex developmental and behavioral programs in vertebrates, such as parturition, testicle descent, bone remodeling, and horn development in sheep[81–86]. Perhaps the (re)interpretation of these programs within a heterochronic perspective could provide insight into the evolution of relaxin-like signaling pathways and their roles in development and disease.

We found that the peripheral peptide hormone, Dilp8, modulates a central neuromotor circuit to switch a motor pattern during the execution of an innate behavior. Different types of extrinsic neuromodulators have been shown to act directly on the central nervous system. Examples are the circulating biogenic amines octopamine and serotonin that regulate posture in lobsters by acting in central circuits[87,88], the gut-microbiota-derived tyramine that modulates an aversive olfactory response of its host, *Caenorhabditis elegans*, by acting on sensory neurons probably after being metabolized into octopamine[89], the peripheral peptides regulating feeding behavior (ghrelin, leptin, insulin, cholecystokinin, peptide YY, and pancreatic polypeptide), which directly or indirectly act on first-order feeding neurons in the hypothalamus and brainstem areas[90–93]. In insects, the ecdysis triggering

hormone (ETH), which is released from inka endocrine cells and acts centrally on abdominal leucokinin (ABLK) neurons, triggers pre-ecdysis behavior. Ecdysis is another insect innate behavior that promotes cuticle shedding. Ecdysis consists of a sequence of three behavioral subunits: pre-ecdysis, ecdysis, and post-ecdysis[94–97]. While both Dilp8 and ETH act directly on the central nervous system, the Dilp8-target neurons are interneurons, while the ABLK neurons send neurites towards the periphery, even though it is not clear if these projections are required for ETH sensing. How exactly Dilp8 transverses the blood-brain barrier to reach the Lgr3-positive interneurons remains to be defined. A similar unresolved issue occurs in the earlier signaling event in the growth control paradigm, where imaginal-disc-derived peripheral Dilp8 acts on brain interneurons[23–28,34,46].

*Drosophila* Lgr3 receptor and its invertebrate orthologs are part of the ancestral group of relaxin receptors together with their vertebrate orthologs, the relaxin family receptors RXFP1 and RXFP2, which respond most specifically to the vertebrate relaxin and insulin-like peptide-3 (INSL3) ligands[46,98]. RXFP1 and RXFP2 regulate innate behaviors and processes such as parturition and testicle-descent, respectively, amongst others. However, the vertebrate relaxin family peptide that most clearly acts as a behavioral neuromodulator is Relaxin-3, which happens to be the ancestral peptide of the vertebrate relaxin family of peptides[98]. It acts via a different receptor class, the RXFP3/4 relaxin receptor family, which is not found in invertebrates. Relaxin-3 acts as an arousal transmitter that works by altering hippocampal theta rhythms and associated learning and memory[99,100]. While Dilp8 acts extrinsically on the CNS via an RXFP1/2-family receptor, and relaxin-3 acts intrinsically via an RXFP3/4-family receptor[99,100], it is clear that neuromodulatory function is an ancestral function of the highly conserved relaxin family of peptides in animals.

## Methods

**Drosophila husbandry and stocks.** *Drosophila virilis* (15010-1051.118 from The National Drosophila Species Stock Center) was a gift from N. Frankel. All other *Drosophila* stocks were *Drosophila melanogaster*. UAS-dilp8 and UAS-dilp8$^{C150A}$ were previously described[24]. Lgr3$^{ag1}$, Lgr3$^{ag2}$, sfGFP::Lgr3$^{ag5}$, and UAS-Lgr3 were previously described[26]. tub-dilp8 (ref. [27]) was a gift from M. Dominguez. Feb36-GAL4 (from C. Thummel)[101]. w; phm-GAL4/TM6Tb, and y w;P0206-GAL4 (ref. [102]) were gifts from C. Mirth. Act88F-GAL4 (ref. [103]) was a gift from F. Schnorrer. ppl-GAL4 (ref. [104]) and dilp8$^{KO}$ (ref. [40]) were gifts from P. Leopold. A58-GAL4 was a gift from M. Galko (ref. [105]). nSyb-GAL4 (III)[106] was a gift from R. Teodoro. UAS–Rho1-IR(1) (VDRC 12734), UAS-Rho1-IR(2) (BL27727 y$^1$ v$^1$; P{y[+t7.7] v[+t1.8]=TRiP.JF02809}attP2), and forkhead-GAL4 (BL78060 w[*]; P{w[+mC]=fkh-GAL4.H]3) were a gift from M. Melani. y$^1$ w$^{67c23}$; P{CaryP}attP40;; was obtained from the Champalimaud Foundation Injection Facility (a gift from N. Perrimon). UAS-dilp8-IR (v102604)({KK112161}VIE-260B)) and UAS-EcR-IR (w[1118]; P{GD1428}v37059) (ref. [107]) were obtained from the Vienna Drosophila Resource Center (VDRC). vas-int; attP40 (Stock 13-20), full genotype: y w M(eGFP, vas-int, dmRFP)ZH-2A; P{CaryP}attP40 (ref. [108]) was obtained from Fly Facility, Department of Genetics, University of Cambridge.

The following stocks were obtained from the Bloomington *Drosophila* Stock Center at Indiana University:

BL33079 y$^1$ w*; Mi{MIC}Ilp8MI00727
BL54591 y$^1$ M{w[+mC]=nos-Cas9.P}ZH-2A w*
BL58986 P{ry[+t7.2]=hsFLP}12, y$^1$ w*; P{y[+t7.7] w[+mC]=UAS-Cas9.P2}attP2/TM6B, Tb$^1$
BL49275 w$^{1118}$; P{y[+t7.7] w[+mC]=GMR17G11-GAL4}attP2
BL48786 w$^{1118}$; P{y[+t7.7] w[+mC]=GMR17H01-GAL4}attP2
BL48806 w$^{1118}$; P{y[+t7.7] w[+mC]=GMR18C07-GAL4}attP2
BL48791 w$^{1118}$; P{y[+t7.7] w[+mC]=GMR18A01-GAL4}attP2
BL48840 w$^{1118}$; P{y[+t7,7] w[+mC]=GMR19B09-GAL4}attP2
BL39171 w$^{1118}$; P{y[+t7.7] w[+mC]=GMR57C10-GAL4}attP2
BL50395 w$^{1118}$; P{y[+t7.7] w[+mC]=GMR48H10-GAL4}attP2
BL27390 y$^1$ w*; P{w[+mC]=GAL4-Mef2.R}3
BL44277 w$^{1118}$; P{y[+t7.7] w[+mC]=13XLexAop2-IVS-GCaMP6f-p10}su(Hw)attP5
BL5885 w*; P{w[+m*]=Sgs3-GFP}3
BL32219 w*; P{10XUAS-IVS-mCD8::RFP}attP40
BL55819 w$^{1118}$; P{y[+t7.7] w[+mC]=8XLexAop2-FLPL}attP2
BL38879 P{w[+mC]=alphaTub84B(FRT.GAL80)}1, w*; Bl$^1$/CyO; TM2/TM6B, Tb$^1$
BL32199 w$^{1118}$; P{10XUAS-IVS-myr::GFP}su(Hw)attP5
BL5138 y$^1$ w*; P{w[+mC]=tubP-GAL4}LL7/TM3, Sb$^1$ Ser$^1$
BL7016 P{w[+mC]=tubP-GAL80[ts]}Sxl[9], w[*]/FM7c
BL80436 y$^1$ v$^1$; P{y[+t7.7] v[+t1.8]=TRiP.HMS06016}attP40
BL6871 w{1118}; P{w[+mC]=MsrA-GAL4.657}TP1-1 (Eip71CD-GAL4)
B28281 w*; P{w[+mC]=UAS-RedStinger}6, P{w[+mC]=UAS-FLP.Exel}3, P{w[+mC]=Ubi-p63E(FRT.STOP)Stinger}15F2 (G-TRACE stock[109])

All other stocks were generated in this study as described below. Stocks are maintained at low densities at 18 °C in a 12-h light/dark cycle.

**Ceratitis husbandry and sample collection.** The *C. capitata* culture was kindly provided by Dr. A. Jessup and was maintained on a diet of sugar and hydrolyzed yeast protein for the adults and on a *Drosophila* food medium for the larvae. The eggs were collected and placed on the food at room temperature. After three to four days, hatching was confirmed and the bottles containing the larvae were transferred to 25 °C. Approximately 10 days after the egg lay, the larvae started to crawl out of the food at which point the bottles were placed on sawdust with the bottle caps removed. The larvae either crawled or jumped out of the bottle onto the sawdust to start the pupariation process. The larvae were collected from the sawdust and placed in a petri dish. After a period of continued crawling and jumping (wandering stage), the larvae ceased to be active and started to contract into an oblong shape (contraction stage). 1–1.5 h after the start of the contraction, the larvae become externally immobile (white-prepupa stage). For RNA samples, three individuals were collected in 1 ml Trizol with three replicates per time point.

**Puparium aspect ratio, pupariation time measurements, and pupa staging.** Pictures of puparia were taken under a dissecting scope and were analyzed using ImageJ and/or Amscope software. Length was measured from the anteriormost edge of the pupa to the most anterior anal papilla (Supplementary Fig. 10a) Width was measured in the widest part of the middle third of the pupa. Animal sex was not taken into account, because AR is equivalent in males and females, despite female puparia being proportionally longer and wider than male puparia. Pupariation time was measured essentially as previously described[26]. Briefly, females were allowed to lay eggs for 4 h and on the next day, L1 animals were transferred to a vial with regular food at low density (~30 animals per tube) and kept at 25 °C with constant light. The number of animals that reached pupal stage was counted twice a day. Animals were classified as white prepupa by direct observation through the wall of the vial. Immobile larvae were circled with a marker and regularly observed until clear signs of operculum formation were already seen (mainly, the anterior segments begin to flatten and their lateral edges thicken and become straight), but the animal was still white (Supplementary Fig. 10b).

**Immunofluorescence analyses.** CNS or carcass (integument and body wall muscle) of WPP T0 animals were dissected in Schneider Medium (Biowest – cat. # L0207 or Gibco - cat. #21720-024), fixed for 30 min in 4% paraformaldehyde, rinsed with PBS with Triton (0.3%) (PBST), incubated with primary antibody overnight and with fluorescently labeled secondary antibody for 2–24 h in PBST with 1% bovine serum albumin. Samples were washed 3× for 30 min each in PBS after each antibody incubation. Nuclei were counterstained with DAPI (Sigma) and tissues were mounted in Fluoromount-G (Southern Biotech). Antibodies used were: rabbit anti-GFP 1:200 (Life technologies, A11122) and mouse 7G10 anti-Fasciclin III (anti-FasIII) 1:50 [developed by C. Goodman (University of California, Berkeley) and obtained from the Developmental Studies Hybridoma Bank (DSHB), created by the NICHD of the NIH and maintained at The University of Iowa, Department of Biology, Iowa City, IA 52242][110]. Images were obtained with a Zeiss LSM 710 Confocal Microscope and images were analyzed using FIJI software[111]. Typically 5–10 CNSs were mounted for observation and 1 representative image per genotype is depicted in figures. CNSs from male and female larvae were scored together.

**General molecular biology.** gDNA was extracted as previously described[26,112]. Briefly, one or two flies were macerated using pellet pestles and homogenized in 100 μl DNA extraction buffer (1 M Tris-HCl at pH 8.2, 0.5 M EDTA, 5 M NaCl). Then, we added 1 μl proteinase K (final concentration of 400 μg/mL), and incubated the mixture at 37 °C for 1 h, followed by 95 °C for 5 min, to inactivate the protease.

RNA was extracted using either the Direct-zol RNA MiniPrep kit (Zymo Research), High Pure RNA Tissue Kit (Roche) or NZY Total RNA isolation kit (NZYtech), following the manufacturer's instructions. The material used for the qRT-PCR experiments described in Figs. 2, 6j, and 7n were obtained from 1-5 staged animals, depending on the experiment, and was macerated using pellet pestles and homogenized in 800 μl of TRI Reagent or NZYol and centrifuged at 12,000 × g for 1 min, to lower tissue debris. After the centrifugation, half volume of absolute ethanol was added to the supernatant and mixed well. Then, the sample was loaded in a binding column of the RNA extraction kit. An extra DNAse treatment (Turbo DNA-free kit, Ambion, Life Technologies) was performed to reduce gDNA contamination. cDNA synthesis was performed using the Maxima First Strand cDNA Synthesis Kit for RT–quantitative PCR (Thermo Scientific) or NZY First-Strand cDNA Synthesis Kit, following manufacturer's instructions.

In situ hybridization probes, PCR, and qRT-PCR primers are described in their respective sections below and in Supplementary Table 1. Briefly, their specificity was tested using Primer BLAST or Primer3. Primers and probes for *Ceratitis capitata* were obtained from InsectBase http://www.insect-genome.com/ [Whole genome assembly of Mediterranean fruit fly (*Ceratitis capitata*) as part of the BCM-HGSC i5k Pilot Project; ref. [113]]. *C. capitata ilp8* (*cilp8*) corresponds to uncharacterized protein LOC101461861 [*Ceratitis capitata*], NCBI Reference Sequence: XP_004525593.1, Gene ID GI: 498965474. *C. capitata Rp49* (*cRp49*) corresponds to LOC101451559 60 S ribosomal protein L32 [*Ceratitis capitata*], NCBI Reference Sequence: XP_004517954.1, Gene ID: 101451559.

**20HE treatment**. *dilp8^ag52* flies were left to lay eggs for 2 h on apple plates. 20 to 30 larvae were transferred to vials with normal food at 48 h after egg laying. Larvae were then collected at 96 h after egg laying, washed in PBS, and the carcass was dissected from the rest of the larva tissue in Schneider Medium (Gibco - cat. #21720-024). Two carcasses were incubated for each treatment in a 24-well dish. The carcasses were incubated in Schneider medium for 1 h with oxygenation by agitation (250 rpm) at room temperature (22–25°C). This timepoint corresponded to the T0 sample (before treatment). The Schneider medium was then replaced with a fresh medium containing 20-hydroxyecdysone (Cayman Chemical cat. #16145) in a final concentration of 5 μM[54] or equivalent volume of vehicle (absolute ethanol) for 3-6 h after which the carcass was frozen in dry ice and stored in -80 °C conditions until processing for qRT-PCR as described above.

**Treatment with mDOPA**. Female flies were left to lay eggs for 6 h on egg-laying plates covered with a thin layer of normal food. Thirty to 35 larvae were transferred to vials with 23 mM α-methyldopa (Hipermet, Lab Raymos) or an equivalent volume of solvent (water) 72 h after egg laying. Pupariation behavior of wandering larvae carrying mhc»GCaMP was assessed in the pupariation monitoring device. The remaining larvae that pupariated in the wall of the vials were used to calculate the AR of the puparium as described above. All the experimental procedures were performed at 25 °C.

**Expression of *dilp8* after the midthird instar transition**. Flies were left to lay eggs for 4 h on egg-laying plates covered with a thin layer of normal food at 18 °C. Thirty to 35 larvae were transferred to vials with normal food 96 h after egg laying (4 d) and maintained at 18 °C until 169 h after egg laying (7 d + 1 h), when they were shifted to 30 °C. Wandering larvae were transferred to the pupariation monitoring device 4–5 h later and videos were filmed at 30 °C. As a control condition, animals with the same genotypes were bred and filmed at 18 °C.

Preliminary experiments showed that expression of *dilp8* after 4 and 6 d of development at 18 °C induce a ~60-h and ~40-h delay in pupariation, respectively. On the contrary, when larvae were switched to 30 °C 7 d after egg laying, no delay in development was observed. Instead, *dilp8* mutants, like *Lgr3* mutants, pupariate ~4 h earlier than WT animals, as expected from their non-rescued phenotypes as regards pupariation timing control by PIL/GCL neurons[23–26]. This also suggested that the endogenous role of *dilp8* in pupariation timing control in the absence of induced imaginal disc tissue growth aberrations is played prior to the midthird instar transition.

**Germline CRISPR-Cas9 generation of *dilp8* alleles**. To test if the *Lgr3* puparium morphogenesis phenotype was related to the function Lgr3 plays as a receptor for Dilp8, we first quantified AR in animals carrying a hypomorphic *dilp8* allele, *dilp8^MI00727* (an eGFP enhancer trap[24,114]) or two different RNAi lines against *dilp8* [*dilp8-IR^KK* (refs.[23,24,41]) or *dilp8-IR^TRIP* (ref.[115])], ubiquitously-expressed under the control of different drivers, but did not observe a consistent phenotype (Supplementary Fig. 1b–g). As it is possible that none of the hypomorphic conditions removed sufficiently enough of Dilp8 activity to affect puparium AR, we generated *dilp8* mutants using CRISPR/Cas9-mediated directed mutagenesis[47,48]. For this, we used a single specific guide RNA (gRNA) against *dilp8* (*dilp8^gRNA1*) to guide the germline Cas9 endonuclease activity (*nos-Cas9.P*)[48] to the 3' end of the *dilp8* locus, which encodes essential cysteines that are critical for Dilp8 activity[24]. We obtained 5 indel alleles of *dilp8* (*dilp8^ag50*, *dilp8^ag51*, *dilp8^ag53*, *dilp8^ag54*, and *dilp8^ag55*) all of which are predicted to severely disrupt Dilp8 activity (Supplementary Fig. 1a). One of the indels, *dilp8^ag50*, is a 570-bp deletion + 5-bp insertion that removes approximately half of the sequence coding for the Dilp8 carboxy-terminus (Supplementary Fig. 1a). We also kept a background control allele *dilp8^ag52*, where the *dilp8* sequence was intact (Supplementary Fig. 1a).

Technically, plasmid pU6-BbsI-chiRNA-dilp8_gRNA1 was generated by cloning the annealed primers #200_DILP8-GuideRNA_1_F CTTCGCACTGGTTTAGACAGCAGT and #201_DILP8-GuideRNA_1_R AAACACTGCTGTCTAAACCAGTGC into BbsI-digested pU6-BbsI-chiRNA [a gift from Melissa Harrison & Kate O'Connor-Giles & Jill Wildonger (Addgene plasmid # 45946; http://n2t.net/addgene:45946; RRID:Addgene_45946)], as previously described[26,47]. pU6-BbsI-chiRNA-dilp8_gRNA1 was injected into BL54591 *y^1 M{[+mC]=nos-Cas9.P}ZH-2A w** flies, after which the mutagenized 3rd chromosome was isolated by crossing to *w^1118; If/CyO; MKRS/TM6B* flies, and then to *w^1118;; MKRS/TM6B* flies to select *w^1118;; dilp8*/TM6B* animals. Candidate indels were detected by PCR using non-*TM6B* homozygous *w^1118;; dilp8** animals

using primers: #107_dilp8_salto_exon2_R CAGTTGCATATGTGCCGCTGGA with primer #200 above. All recovered *dilp8* alleles were homozygous viable.

**Tissue-specific CRISPR-Cas9 of *dilp8***. To genetically test if the cuticle epidermis is the primary source of Dilp8 activity that signals to Lgr3 in *R18A01* neurons to mediate proper puparium morphogenesis, we attempted to carry out tissue-specific CRISPR-Cas9 experiments using a *UAS-Cas9.P2* transgene and the same *dilp8* guideRNA sequence used for germline CRISPR-Cas9[48,116] (generating the stock *pCFD6-dilp8^gRNA1*, described below) to knockout *dilp8* in cuticle epidermis cells. Unfortunately, these experiments were hindered by the fact that the cuticle epidermis seems to be particularly sensitive to toxicity effects of the Cas9.P2 endonuclease[48,117]. Specifically, Cas9.P2 expression alone caused phenotypes that are epistatic to the puparium AR phenotype, precluding specific conclusions about the tissue-specific requirement for *dilp8* in epidermal cells of the cuticle (Supplementary Fig. 3a, b).

**Generation of *pCFD6-dilp8^gRNA1* stock**. To generate *w^1118; {pCFD6-dilp8^gRNA1} attp40;* transgenic animals, the same primary gRNA sequence used for germline CRISPR-Cas9 experiments described above was adapted and cloned into BbsI-digested *pCFD6* plasmid [a gift from Simon Bullock (Addgene plasmid # 73915; http://n2t.net/addgene:73915; RRID:Addgene_73915)[116] using a primer annealing strategy with primers #681_DILP8-GuideRNA_1_F-ALT TGCAGCACTGGTT-TAGACAGCAGT and #201_DILP8-GuideRNA_1_R, AAA-CACTGCTGTCTAAACCAGTGC. to allow *dilp8^gRNA1* expression under the control of *UAS* sequences. *pCFD6-dilp8^gRNA1* was then injected into the *Drosophila* stock *w M(eGFP, vas-int, dmRFP)ZH-2A; P{CaryP}attP40* for PhiC31 transgenesis[108] (from the Champalimaud Foundation Drosophila Injection Facility). Transgenic animals were selected by eye color and balanced against *w^1118; If/ CyO; MKRS/TM6B*.

**Generation of the *mhc-LHV_2* stock**. In order to generate the *mhc-LHV_2* stock, we amplified the *LHV_2* ORF (a gift from Ryohei Yagi and Konrad Basler)[62] using primers D-TOPO_LHV2_F CACCAAGCCTCCTGAAAGATG and D-TOPO_LHV2_R AATGTATCTTATCATGTCTAGAT. The ORF was then inserted into an entry vector using pENTR Directional TOPO cloning (Invitrogen) followed by Gateway cloning reaction into a *mhc* destination plasmid (mhc-Gateway, a gift from Brian McCabe). Transgenic lines were generated by standard P-element-mediated transformation procedures in a *yw* background. Insertions on the 2nd and 3rd chromosome were balanced against *w^1118; If/CyO; MKRS/TM6B*.

**Generation of stock *R18A01-LexA***. To generate *y^1, w^67c23; P{BP_R18A01-LexA:: p65Uw}attP40/CyO* (*R18A01-LexA*), we made the plasmid pBP_R18A01_LexA:: p65Uw by amplifying the *R18A01* regulatory element region[49,50] using primers #477_R18A01_Left_primer GCTTAGCCAGATTGTTGGATGCCTG and #478_R18A01_Right_primer GCGTTATGAGGTTGTGCTGCAGATC and cloning it into pBPLexA::p65Uw [a gift from Gerald Rubin (Addgene plasmid # 26231; http://n2t.net/addgene:26231; RRID:Addgene_26231)[61] using standard Gateway cloning procedures. The transgenic animals were generated by standard PhiC31 transformation[118,119] by injecting the final plasmid into the stock *y^1 w^67c23; P (CaryP)attP40;;* (ref. [108]) (from the Champalimaud Foundation Drosophila Injection Facility). Transgenics were selected by eye color and balanced against *w^1118; If/ CyO; MKRS/TM6B*.

**Generation of stock *13xLexAop2-GAL4v-VP48***. To generate *w^1118; {13xLexAop2-GAL4v-VP48}attp40;;* we made the plasmid pJFRC19-13XLexAop2-V5-GAL4v-VP48-OLLAS by substituting the XhoI-XbaI myr::GFP fragment from plasmid pJFRC19-13XLexAop2-IVS-myr::GFP [a gift from Gerald Rubin (Addgene plasmid # 26224; http://n2t.net/addgene:26224; RRID:Addgene_26224)[61] for a XhoI-XbaI V5-GAL4v-VP48-OLLAS fragment from pUC57-V5-GAL4v-VP48-OLLAS. The latter plasmid was generated by placing the following de novo synthesized V5-GAL4v-VP48-OLLAS (GAL4-VP48) sequence (Genscript) into EcoRV-digested pUC57. This codon-optimized and functional GAL4 transcriptional activator variant was used here due to the convenient restriction sites and its full characterization will be published elsewhere.

*V5-GAL4V-VP48-OLLAS (BLUE, CAPITALIZED AND ITALICS)* + flanking restriction sites (lowercase) placed into pUC57

ggtaccgaattcctcgaggccaccATGGGCAAGCCCATCCCGAACCCACTGCTGG GCCTGGATTCCACCAAGCTGCTGAGCTCCATCGAGCAGGCCTGCGACA TCTGCCGCCTGAAGAAGCTGAAGTGCTCGAAGGAGAAGCCCAAGTGC GCCAAGTGCCTGAAGAACAATTGGGAGTGCCGCTACTCCCCCAAGACC AAGCGCTCCGCCGCTGACCCGCGCCCACCTGACCGAGGTGGAGACCGC CTGGAGCGCCTGGAGCAGCTGTTCCTGCTGATCTTCCCGCGCGAGGATC TGGACATGATCCTGAAGATGGATTCCCTGCAAGACATCAAGGCCCTGC TGACCGGCCTGTTCGTGCAGGATAACGTGAATAAGGATGCCGTGACCG ACCGCCTGGCCTCCGTGGAGACGGACATGCCACTGACCCTGCGCCAGC ACCGCATCTCGGCCACCAGCAGCAGCGAGGAGTCGAGCAACAAGGGC CAGCGCCAGCTGACCGTGAGCGAGTTCGAGTGCGAGTTCCTGACCCG CTCCGGCTACAGCAGCAGCGATGTGCGCGGCAAGTGCTGGGAGCCCAC

CGACGCCCTGGATGACTTCGATCTGGACATGCTGCCAGCCGATGCCCTG
GATGATTTTGATCTGGACATGCTGCCCGCCGACGCCCTGGATGATTTTG
ATCTGGACATGTTACCAGGCTCGGGATTCGCCAATGAGCTGGGACCACG
CCTGATGGGCAAGTAAgcggccgcggatcctctaga.

The transgenic animals were generated by standard PhiC31 transformation[118,119] by injecting the final plasmid into the stock $y^1$ $w^{67c23}$; $P$ (CaryP)attP40;; (ref. [108]) (from the Champalimaud Foundation Drosophila Injection Facility). Transgenics were selected by eye color and balanced against $w^{1118}$; If/CyO; MKRS/TM6B.

**Generation of the intersection stock R18A01 ∩ R48H10 >**. To generate the R18A01 ∩ R48H10 > intersection stock, we first generated a R180A1-LexA::p65 (R18A01») line (Supplementary Fig. 9b) as described above. In order for the intersection to work, R18A01» must drive expression in the same cell types and in similar strength as the R18A01 > version. If this is true, then knockdown of Lgr3 by RNAi in R18A01»-positive cells should increase puparium AR. To confirm this, we generated a LexAop2-GAL4-VP48 stock and coupled it to UAS-Lgr3-IR transgene or to a UAS-CD8::GFP transgene. As expected, R18A01»GAL4-VP48 > Lgr3-IR strongly increased puparium AR and R18A01»GAL4-VP48 > CD8::GFP expression pattern was similar to the pattern obtained using R18A01 GAL4 line (Supplementary Fig. 9c, d). We then coupled R18A01» with R48H10 > by adding a LexA-dependent flippase (8x-LexAop2-FLPL) cassette and GAL80 flip-out cassette (alphaTub84B(FRT.GAL80)), where GAL80 is flanked by FRT sites and can thus be removed by flippase recombinase activity[69]. In this way, the R48H10 > activity is blocked in every cell except in those in which the R18A01»FLP flips-out the inhibitory GAL80.

**qRT-PCR**. qRT–PCR experiments were performed as described previously[26]. Briefly, the experiments were performed in a Lightcycler 96 (Roche) using the FastStart Essential DNA Green Master dye and polymerase (Roche). The final volume for each reaction was 10 µl, consisting of 5 µl of dye and polymerase (master mix), 2 µl of cDNA sample (diluted to an estimated 1-10 ng/µl equivalent of RNA) and 3 µl of the specific primer pairs.

The efficiency of the primers for the qPCR was verified by analyzing a standard curve with 3 serial dilutions of gDNA from $w^{1118}$ animals and the production of primer dimer was checked by analyzing the melting curve. Only primers with more than 90% of efficiency and with no primer dimer formation were used. FastStart Essential DNA Green Master (Roche) in a LightCycler® 96 Instrument (Roche) was used for performing the PCR steps. qRT-PCR results were expressed as mRNA levels relative to rp49 (rp49 = 100). Primer pairs used for D. melanogaster genes rp49, dilp8, pale, dib, phm, and E74B[120] and Ceratitis capitata genes rp49 (Ccap_RpL32) and cilp8 are available in Supplementary Table 1.

**In situ hybridization**. The wandering larvae and white prepupae were cut open along the ventral midline and spread on a Sylgard petri dish in PBS. After clearing the epidermis of other tissues, it was fixed for one hour in 4% formaldehyde in PBS. The tissue was washed in PBT (PBS, 0.1% Tween-20), cut into smaller longitudinal pieces, dehydrated in 100% methanol, and stored at −20 °C until use. A 413-bp fragment from cilp8 was amplified by PCR using primers #654 cilp8_probe_fwd TGAGAACACTATTCCTTACATTCTTC and #655 cilp8_probe_rev GAAATCCTCTTCACATTTGTTGT using as template the cDNA obtained by oligodT-reverse transcriptase reaction of mRNA isolated from puparating C. capitata animals. The fragment was cloned into the pGEM plasmid in a 3' to 5' direction to generate plasmid #444 pGEM-Cilp8_probe, which was used for riboprobe transcription. In situ hybridization was carried out as previously described[121,122], with the following modifications: the epidermis were incubated for 3 min in 4 µg/mL proteinase K at 37 °C, the hybridization buffer included heparin instead of glycogen and the hybridization step was carried out at 60 °C. Tissues were mounted in 70% glycerol in PBS and observed under the Leica DM LB2 upright microscope.

**Puparation monitoring device**. The puparation monitoring device consists of a camera connected to a Raspberry Pi microcomputer and a 3D-printed puparation arena illuminated with LEDs.

Puparation arenas were designed using Freecad and exported as.srt files for 3D printing. They consisted of 3 chambers of 32.0 × 12.0 × 2.0 mm, 6 chambers of 10.0 × 12.0 × 2.0 mm in a 3 × 2 array, or 15 chambers of 5.0 × 5.0 × 1.5 mm in a 5 × 3 array. All chambers were connected between them and to the outside by small grooves that were pre-designed or made with a scalpel after printing. The 6-chamber arena also had small chambers connected to each main chamber designed to contain a piece of agar to avoid larval desiccation when videos were recorded in incubators without humidity control. Larvae were filmed individually or in groups of up to six animals, depending on the size of the arena.

3D-printed arenas were placed between two pieces of glass held with metal clips or double-sided adhesive tape and placed in vertical position in front of the camera of a Raspberry Pi at an adaptable focal distance. For larval monitoring under white light, two pieces of 12-V white LED strips, each with 3 LEDs, or 6 flat 5-mm through hole LEDs (5 V, 1400 mcd, 100°) were positioned in front of the arena, above and below the camera. For mhc»GCaMP transgenic larvae monitoring, two

pieces of 12-V blue LED strips, each with 3 LEDs or 6 flat 5-mm through hole LEDs (5 V, 600 mcd, 100°) were used. A green filter was placed ahead of the lens of the camera to block blue light (Rosco Permacolor Dichroic Filter, #5156 Fern Green). The components of the puparation monitoring device were assembled together using LEGO blocks or laser-cut acrylic stands.

Videos were recorded at 800 × 600 or 1330 × 1000 pixel resolution when illuminated with white and blue light, respectively. Up to 24-h long videos split in 5-min files were recorded using raspivid command line tool or a custom modification of the FlyPi Graphical User Interface at 10 fps[123] available in GitHub (https://github.com/AndresGarelli/FlyPi-Puparation)[124].

The typical settings used were:

raspivid -rot 180 -p 1050,100,800,600 -w 800 -h 600 -t 43200000 -fps 10 -b 1000000 -ex snow -sg 30000 -o nameOfFile_%04d.h264 for white light illumination and raspivid -rot 180 -p 0,100,600,450 -w 1333 -h 1000 -t 86400000 -fps 10 -b 1000000 -ex snow -sg 300000 -sn 1 -awb off -awbg 1.3,0.1 -o nameOfFile_%04d. h264 for blue light illumination.

A detailed explanation of each parameter can be found in https://www. raspberrypi.org/documentation/raspbian/applications/camera.md

The original 5-min .h264 video files were concatenated, compressed, and saved in the .mp4 container format using ffmpeg software.

*Larva tracking with ImageJ*. For tracking larval behavior, larvae were individually placed in the 3×2 arena and their movement recorded until puparation. Videos were processed as indicated above and one frame per second was extracted and saved as a.bmp image. Position within the chamber, aspect ratio, and brightness were measured for each individual larva using a custom-written ImageJ macro (available in https://github.com/AndresGarelli/ImageJ-Larva-Tracking-Tool[125], with examples and instructions[126]). The data obtained was exported as a.txt file which was further processed in Excel to calculate the position, speed, total distance traveled, and distance to the final position. Each parameter was calculated as follows:

- **Position:** was obtained using the centroid measurement for the larvae in each area using ImageJ.
- **Distance:** is the size in pixels of the straight line connecting two consecutive positions.
- **Total distance traveled:** is the cumulative distance the larva has traveled expressed in pixels.
- **Speed:** is calculated as the distance traveled in the previous 60 s.
- **Distance to final position:** the size in pixels of the straight line connecting current position with the position were the larva pupariates.

Blue LED lighting is not even across each chamber of the puparation arena. As a consequence, basal mhc»GCaMP-fluorescence signal is dependent on the position of the larvae within the chamber and it varies significantly in wandering larvae. However, once the larvae stop wandering and pre-PMP begins, changes in intensity reflect actual GCaMP fluctuations.

For the analysis of GCaMP fluctuations, the following parameters were calculated:

- **Duration of pre-GSB contractions:** time in seconds during which the fluorescence intensity of GCaMP is above 50% of the difference between the baseline ($F_0$) and maximum ($F_{max}$) values of each peak, where $F_0$ is the minimum value of the preceding 10 s.
- **Duration of post-GSB contractions:** time in seconds during which the fluorescence intensity of GCaMP is above a moving threshold value. A central moving minimum (CMM) was calculated as the minimum fluorescence value of a 40 s window centered on each point and the threshold was typically set 5% above CMM, though in some occasions that percentage was adjusted to improve peak discrimination.
- **Amplitude:** is ($\Delta F/F_0$) where $\Delta F$ is ($F_{max}-F_0$) and $F_{max}$ and $F_0$ are the maximum and minimum fluorescence value of a peak.
- **Period:** is the time between the start of two consecutive peaks of GCaMP.

The four stages of the puparation motor program (pre-GSB, GSB, and post-GSB-1 and 2) were consistently observed in control-type animals, with only small differences in the number, duration and amplitude of contractions between individuals. Representative GCaMP traces reflecting this diversity are shown in Supplementary Fig. 11.

*Criteria to call pre-GSB mhc»GCaMP-fluorescence peaks in mutant animals*. All mutant animals show peaks of GCaMP fluorescence once they stop wandering. These peaks can result from peristaltic waves or whole-body contractions and were considered to be pre-GSB-like contractions only after confirming that the peak corresponded to a contraction of the body. This is certainly a conservative criteria, as those peaks that were not considered to be pre-GSB-like contractions because they were weak by visual inspection could have actually been part of the pre-GSB subprogram.

In mutant animals, the beginning of pre-GSB is determined by the first short-pre-GSB contraction that can be observed, as defined above. Mutant animals do not progress beyond short-pre-GSB stage. Then, the end of pre-GSB is reached when mhc»GCaMP-fluorescence peaks do not lead to body shortening, which

coincides with cuticle sclerotization. Examples of representative GCaMP traces of mutant animals that perform pre-GSB contractions or that lack identifiable generalized body contractions after completion of wandering stage are provided (Supplementary Fig. 12).

**Statistical analyses**. Results and details of all statistical analyses are available in Supplementary Table 2. For all tests, alpha was set at 0.05 a priori. For quantitative data, comparison between multiple conditions were done using ANOVA when samples had normal distribution (Shapiro-Wilk test and equal variance). If these conditions were not met, Kruskal-Wallis One Way Analysis of Variance on Ranks was performed. If the result of these tests was statistically significant, then Dunn's, Holm-Sidak, Student-Neuwan-Keuls or Bonferroni post-hoc tests were applied for the assessment of statistical significance of pairwise comparisons.

Comparison between two conditions were done using one-tailed unpaired (unless noted otherwise) Student's *t*-test, when samples had normal distribution and equal variance. Alternatively, a Mann-Whitney Rank Sum Test or Wilcoxon signed rank test (for paired samples) was performed. Statistically-significantly different comparisons were denoted with an asterisk.

For frequency (binary) data, multiple binomial tests were performed and a Bonferonni correction was applied (alpha/number of pairwise comparisons) to assess statistical significance compared to the following expected distributions: 0.01 for anterior retraction defects, 0.05 for pupal death (viability) data, and the conservative value of 0.005 was considered for GSB failure, based on observations of 376 controls in different backgrounds where only 1 failed to perform a detectable GSB. For specific 2 × 2 comparisons, the Fisher Exact Test was used with an alpha = 0.05.

Results from multiple comparisons were presented, except were explicitly denoted otherwise, with a letter scheme where conditions or genotypes sharing the same letter were not statistically significantly different.

Statistical analyses were performed either using SigmaPlot package or at https://www.socscistatistics.com/ (for binomial tests).

**Reporting summary**. Further information on research design is available in the Nature Research Reporting Summary linked to this article.

## Data availability

All relevant data are available from the authors. Source data are provided with this paper. Data used in Supplementary Fig. 2a, d were obtained from microarray data[54] deposited in the National Center for Biotechnology Information Gene Expression Omnibus website, with accession numbers as follows: GSE3057 for Supplementary Fig. 2a, d and GSE3069 for Supplementary Fig. 2c, f. Data used in Supplementary Fig. 2b, e were obtained from microarray data[53] available from https://doi.org/10.1016/S1534-5807(03)00192-8. Source data are provided with this paper.

## Code availability

Custom code developed to acquire videos using the pupariation monitoring device is available in https://github.com/AndresGarelli/FlyPi-Pupariation[124] and ImageJ macros in https://github.com/AndresGarelli/ImageJ-Larva-Tracking-Tool[125], with examples and instructions[126].

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

## Acknowledgements

We thank Drs. Carlos Ribeiro, Christen Mirth, Elio Sucena, Filip Port, Frank Schnorrer, Julien Colombani, Maria Dominguez, Maria Luisa Vasconcelos, Pierre Leopold, Simon Bullock, Rita Teodoro, Gerald Rubin, Melissa Harrison, Kate O'Connor-Giles, Jill Wildonger, Mariana Melani, Pablo Wappner, and Christian Wegener for fly stocks and reagents. We thank Ryohei Yagi and Konrad Basler for the $LHV_2$ plasmid and Brain McCabe for the *mhc*-Gateway destination plasmid. We thank Carlos Ribeiro and Dennis Goldschmidt for help in designing and constructing one of the pupariation arenas and Mariana Melani, Pablo Wappner, Arash Bashirullah, and Filip Port for sharing resources and unpublished data. We thank Arash Bashirullah, Fillip Port, and Carlos Ribeiro for discussions and/or comments on the manuscript, and Jim Truman for discussions on Fraenkel's pupariation factors. Stocks obtained from the Bloomington Drosophila Stock Center (NIH P40OD018537) were used in this study. Work in the Integrative Biomedicine Laboratory was supported by the European Commission FP7 (PCIG13-GA-2013-618847), by the FCT (IF/00022/2012; Congento LISBOA-01-0145-FEDER-022170, co-financed by FCT/Lisboa2020; UID/Multi/04462/2019; PTDC/BEXBCM/1370/2014; PTDC/MED-NEU/30753/2017; PTDC/BIA-BID/31071/2017; FCT SFRH/BPD/94112/2013; SFRH/BD/94931/2013), the MIT Portugal Program (MIT-EXPL/BIO/0097/2017), and FAPESP (16/09659-3, 16/10342-4, and 17/17904-0). AG is a CONICET researcher, YV holds a CONICET postdoctoral fellowship and FPS and MJD hold a PhD fellowship from CONICET. Work in the Garelli lab was supported by ANPCyT (Agencia Nacional para la Promoción de la Ciencia y la Tecnología, PICT 2014-2900 and PICT 2017-0254) and CONICET (PIP11220150100182CO).

## Author contributions

F.H., Y.V., J.P., M.F., M.A., F.H.P.S., M.J.D., and A.G. performed genetic, phenotypic, molecular, and behavioral experiments. F.H., Y.V., A.P.C., A.M., R.D.M., A.G., and A.M.G. generated and characterized mutants. K.T., F.H., and G.A.C. performed Ceratitis experiments. F.H., Y.V., C.B., M.F., J.M., and A.G. performed qRT-PCRs. F.V. contributed to genetic and phenotypic characterization; J.P., M.F., M.K., M.J.D., A.G., and A.M.G. built pupariation arenas and contributed to behavioral analyses under supervision of A.G.; M.K. and A.G. wrote scripts; C.S.M. developed critical reagents. F.H., T.T.T., A.G., and A.M.G. supervised the work. A.G. and A.M.G. contributed to image and data analyses and wrote the manuscript with the help of all authors.

## Competing interests

The authors declare no competing interests.
