## [Peer Review File · Nature Communications]

Reviewer #1 (Remarks to the Author):

In the manuscript “Innate Behavior Sequence Progression by Peptide-Mediated Interorgan Crosstalk, the authors describe a new activity for the developmental checkpoint hormone Dilp8 and its receptor Lgr3. Dilp8/Lgr3 signaling has previously been shown to regulate 1) the duration of larval development in response to tissue regeneration and 2) growth coordination between damaged and undamaged tissues, and 3) symmetric growth between developing left and right wings. These pathways all involve release of Dilp8 by imaginal discs (the larval precursors to adult tissues) which then signals through Lgr3 expressed in specific neurons in the brain as well as through Lgr3 expressed in the Prothoracic gland – a major source of ecdysone synthesis. In this manuscript, the authors demonstrate that prepupal behaviors, muscle contractions that produce the correct pupal shape, and glue secretion and spreading, which help the pupae adhere to surfaces, are regulated by the activity of Dilp8 and a distinct set of Lgr3-expressing neurons in the brain.

This is a very exciting paper for those of us in the field, as it demonstrates that Dilp8/Lgr3 signaling can play important roles beyond imaginal disc growth homeostasis, and develops some tools for examining Dilp8/Lgr3 regulation of pupariation behavior. For the broader research community, it also demonstrates a mechanism by which systemic signals that globally coordinate developmental progression can also act to coordinate that developmental progression with developmentally-cued behaviors. The experiments are well-designed, and for the most part very clearly presented. In general, I am extremely enthusiastic about the publication of this paper. However, I have found a few issues that I think should be addressed in advance of final publication which I describe below:

1) An important part of the authors’ model is that Dilp8, which is expressed in the epidermis in response to 20-hydroxyecdysone at puparium formation, signals to Lgr3-expressing neurons in the CNS to control pupariation behaviors. The authors explain that they were not able to generate a tissue-specific dilp8 knockout or knockdown to test this directly. However, they argue that their experiment rescuing AR in a dilp8 mutant by using a ubiquitously-expressed Dilp8 (*tub>dilp8*) expressed after MIT, supports their model (Fig. 2I). While it does show that dilp8 expression after MIT is sufficient to rescue AR, this experiment doesn’t address the important spatial part of their model. I would suggest that the authors do a similar experiment and see if this rescue can happen using an epidermal driver to express Dilp8 specifically in the epidermal cells. This rescue would then tell us that the Dilp8 expressed in the epidermis can actually rescue AR, through communicating with the 18A01 neurons. This would also be useful for their conclusion that this same signaling pathway, from epidermis to 18A01 neurons, is required for GSB in figure 4H.

2) The characterization of PMP is quite interesting and potentially valuable. However, for it to be most useful for researchers who want to follow up on these experiments, it needs to be clearer how the authors are defining the beginning and end of each period. For instance, even in WT animals (figure 3C) it is not obvious how the transition from post-GSB1 to post-GSB2 is defined. The authors say that they are “characterized by different total mhc>>GCaMP-fluorescence fluctuation patterns” (page 15, line 13), but I couldn’t find where the authors had clearly defined these different patterns or how they used them to assign the boundary. It would be useful for the authors to give us a more complete description of how those transition points are set. This becomes especially important when interpreting mutant phenotypes. For instance, how are the authors determining the duration of pre-GSB in dilp8 mutants (figure 3J)? Why are those patterns that they highlight in green clearly pre-GSB in nature? Alternately, looking at the effects of mDopa on dilp8 mutants (figure S5 D and E), it isn’t clear how those stage assignments are being made. How is the beginning of pre-GSB defined?

the end?

Another difficulty that arises from the characterization of PMP is that it is unclear what the phenotypic variation is within experimental replicates. For the *dilp8* mutants, the authors show us two different traces (presumably two different animals, figure 3J and K) that look extremely different. Do those represent the range of phenotypes? are they representative samples? Is this variability seen in WT controls as well? What about other mutants? It would be valuable if the authors could provide some information that could help readers appreciate the ranges of variation, both for evaluating the findings as well as to inform future experiments who would like to extend these studies.

3) The mDopa experiments have some issues with interpretation, and I think this part of the model is the weakest. For figure 5E, the authors demonstrate that mDopa treatment reduces puparium AR in *dilp8* mutants. They interpret this result as demonstrating that *dilp8* mutants don't achieve proper AR due to excess sclerotization produced by loss of *dilp8* signaling. However, the changes produced in AR by mDopa are the same in *dilp8* mutants and *dilp8+* controls. The correct interpretation of this experiment is that the effects on AR from *dilp8* and mDopa appear to function completely independently. Another way to put this is that the change in AR phenotype produced by loss of *dilp8* or mDopa treatment produce exactly the same change regardless of the other perturbation. This would suggest that mDopa and *dilp8* alter AR through independent mechanisms. The *Lgr3* data (figure 5F) is more compelling that there may be an interaction between *Lgr3* and mDopa, but the effect of the *Lgr3* mutant on AR is much stronger as well. It seems like the interpretation suggested by the data is that *Lgr3* may have a *dilp8*-independent effect on Dopa-mediated sclerotization. I think the authors should re-evaluate this data, address these interpretations and reflect them accordingly in their model.

4) Interpretations of GSB with the R18A01 driver are difficult and should be avoided. For instance, in 6I it is not clear whether the loss of GSB is due to the intersectional expression of *Lgr3-IR*, or the presence of the R18A01 driver.

Minor issues:

5) An initial characterization of the *Lgr3* Cis-regulatory regions would be valuable. Through much of the first half of the paper the authors describe the "two separate populations of neurons marked by R19B09 and R18A01" (Page 4 line 22-23) regulate larval duration and puparium morphogenesis separately, but didn't show these distinct populations (in Fig. 6 they show the R48H10 expression pattern, which seems to also influence puparium morphogenesis. It is also important to acknowledge that subsets of regulatory regions may also produce neomorphic patterns of expression, as regulatory enhancers may be separated from regulatory suppressors.

6) Not really clear to me how the authors conclude that "*dilp8* is a direct target of 20HE-dependent signaling in the carcass". (page 7 lines 20-21). Is this based on the timing of the result? are there EcR binding sites in *Dilp8*? Or are they saying that *Dilp8* responds to ecdysone signaling and this could either be through direct binding of EcR to the *Dilp8* regulatory regions, or through indirect transcriptional regulation? I would request that the authors clarify their interpretation of this experiment.

7) The authors describe how Dilp8/Lgr3 signaling regulate a “thresholded system” or a “thresholded morphogenetic mechanism” (Page 10, line 5 and page 12, lines 6-10). It is not clear to me what the authors exactly mean by this and it would be helpful if they could explain this concept more clearly.

8) For the evaluation of pre-PMP locomotion patterns in Lgr3 mutants (extended data figure 3), it is not clear that there are enough data points for each condition to evaluate the differences between the conditions. More datapoints would be helpful here.

9) On page 10, lines 28-30 are confusing. Replace “posteriorly to” with “after” or “following” and insert “likely occur during” in place of “likely”

10) Page 15, line 28: “later” should be “latter”

11) Figure 5 legend (page 18, lines 15 and 16), J and K legends are mislabeled

12) For Figure 5J it is not clear whether the mRNA was isolated from the whole animal or just the carcass.

In summary, I think this will be a very valuable paper to the field. After the authors address the issues detailed above, I would strongly support its publication in Nature Communications.

Reviewer #2 (Remarks to the Author):

In this article, Heredia et al investigated the role of the relaxin-like Dilp8-Lgr3 pathway in puparium morphogenesis and further characterized pupariating behavior by describing the sequence of events taking place during pupariation motor program (PMP). By combining long-term live imaging and quantitative image analyses with very sophisticated genetic experiments, the authors found that ecdysone-dependent Dilp8 expression in epidermal cells signals to a cluster of 6 Lgr3-positive interneurons located in the VNC to promote PMP progression.

This is a well-designed and executed study that provides novel insights into the role of inter-organ crosstalk in controlling an innate behavior. The article is very well written, yet some description of the different PMP behavioral subunits were quite long and confusing. Nevertheless, I have only two main points to strengthen the key findings, and some other minor issues.

Major points,

1- Dilp8, produced by damaged imaginal discs, has been shown to inhibit ecdysone biosynthesis, and increased levels of 20-hydroxyecdysone in late L3 larvae is known to trigger pupariation. Thus to strengthen the proposal that Dilp8 is required downstream of 20HE to promote puparium morphogenesis, it would be important to show that ecdysone biosynthesis and/or signaling is not affected when interfering with dilp8/Lgr3 pathway.

2- It would be important to fully demonstrate the source of Dilp8 is the epidermis, and to circumvent technical limitations in knocking-down dilp8 expression specifically in epidermal cells, the authors

could use *da>EcRRNAi* to suppress Dilp8 expression in these cells (as shown in Extended Data Figure 2C) and measure AR.

This experiment would also answer whether ecdysone signaling in epidermal cells is required for proper pupariation.

3- What are the consequences of pupariation defects for animal fitness or survival?. For instance, *dilp8ag51* and *ag55* show similar AR defects (Fig. 1F) but very different lethality (Extended Fig. 2K). The authors should include a more direct discussion on this issue.

Minor points,

- Pag. 9, line 20: Subheading: "Proper anterior retraction requires the Dilp8-Lgr3 pathway and is essential for survival" would be more appropriated?

- Pag. 10, line 4: "the peak can be detected as soon as 1.0 h after WPP T0 ("T30", Fig. 2A)". I think it should say "T60".

- Pag. 11, line 16-18: "After 2-4 s, large ventral intersegmental (longitudinal) muscles...., a process that helps extrude the anterior spiracles (Fig. 3I, Supplementary Video 2)." I don't think this is the proper video.

- Fig. 2A: miss-alignment between letters and pupal images.

- Fig. 4G: the authors show that *R18A01>Lgr3* rescues AR, but only partially rescues GSB in *Lgr3ag1* animals. Does it mean that GSB is not required for proper puparium morphogenesis (AR)?

- Fig. 5B: why the effect of Dilp8 overexpression was analyzed in a *dilp8* mutant background? I would have anticipated to see this experiment in wild-type animals.

- Extended Data Figure 1A: what do red boxes represent?

- Extended Data Figure 2G: *UAS-dilp8gRNA* should be changed (as far as I understand it is not a *UAS* line).

Reviewer #3 (Remarks to the Author):

This manuscript outlines an impressive study that demonstrates that *dilp8*, a *Drosophila* relaxin-like peptide, plays a key role in the formation of the puparium, which is necessary for the insect to complete metamorphosis. Not only is this an elegant genetic study that outlines the role of *dilp8* in puparium formation, the role of the *dilp8* receptor in this same process, but the authors beautifully characterise the behavioural processes involved in the formation of the puparium. Importantly, the authors go the pains of narrowing down the cells responsible for the behaviours involved in puparium formation. This manuscript represents an enormous amount of work, and I think the authors are to be commended on the quality and rigour of their experimental paradigms. I have a few relatively minor comments that I feel would improve the readability of this manuscript.

In section 10, page 5, I'm not sure what you mean when you say you can manipulate imaginal disc derived Dilp8 using the cis-regulatory module *R19B09*? What is this construct? Is this part of the regulatory region of *Lgr3*, or is it just a construct that is expressed in the same neurons as *Lgr3*? Figure 1I suggests this construct contains regulatory elements specific to *Lgr3*, presumably the source of Dilp8 is not relevant at the level of an enhancer element. I think this section needs some careful rephrasing to avoid confusion.

In figure 2G, it might be worth adding "exogenous Dilp8" to the diagram of the *dilp8* *-/-* larva in which you overexpress *dilp8*.

Section 25, page 7: What is the MIT ecdysone peak? Are you talking about the peak of ecdysone that induces glue production (Sgs3) in the salivary glands?

The plots in figure 4 are misleading. If your Y axis is % of animals performing a behaviour, why have a Yes/No legend for a binary trait? Isn't it enough to see that 0% of the *dilp8*^{-/-} and *lgr3*^{-/-} animals perform the behaviour? Also, I'm not seeing the "same blue letters" indicated in the legend or the binomial statistics.

I didn't understand the mDOPA experiments in section 20-30 on page 19. I think it would be helpful to explain the difference between sclerotization and melanization, and how you expect each to affect the behaviours associated with pupariation, here.

I'm not so convinced of the role of *dilp8*'s role in heterochrony. Yes, it's true that heterochrony is defined as a change in the timing of the progression of development, but the heterochronic genes in *C. elegans* cause cells to skip whole stage or else remain stuck in the cell divisions of a single stage. It seems to me that *dilp8* is instead important in the correct implementation of a developmental transition. This is decidedly different from skipping a stage (or failing to progress beyond a life stage).

Minor comments:

- Are the AR, CRM, MIT, PMP, and GSB acronyms necessary? I find they disrupt the flow of the text because I have to remind myself what they mean.
- To me, carcass implies the remains of dead animals. I wonder if body wall or integument and body wall muscle would be more accurate?

REVIEWER COMMENTS in black italics

Authors' responses in blue

Reviewer #1 (Remarks to the Author):

In the manuscript "Innate Behavior Sequence Progression by Peptide-Mediated Interorgan Crosstalk, the authors describe a new activity for the developmental checkpoint hormone Dilp8 and its receptor Lgr3. Dilp8/Lgr3 signaling has previously been shown to regulate 1) the duration of larval development in response to tissue regeneration and 2) growth coordination between damaged and undamaged tissues, and 3) symmetric growth between developing left and right wings. These pathways all involve release of Dilp8 by imaginal discs (the larval precursors to adult tissues) which then signals through Lgr3 expressed in specific neurons in the brain as well as through Lgr3 expressed in the Prothoracic gland – a major source of ecdysone synthesis. In this manuscript, the authors demonstrate that prepupal behaviors, muscle contractions that produce the correct pupal shape, and glue secretion and spreading, which help the pupae adhere to surfaces, are regulated by the activity of Dilp8 and a distinct set of Lgr3-expressing neurons in the brain.

This is a very exciting paper for those of us in the field, as it demonstrates that Dilp8/Lgr3 signaling can play important roles beyond imaginal disc growth homeostasis and develops some tools for examining Dilp8/Lgr3 regulation of pupariation behavior. For the broader research community, it also demonstrates a mechanism by which systemic signals that globally coordinate developmental progression can also act to coordinate that developmental progression with developmentally-cued behaviors. The experiments are well-designed, and for the most part very clearly presented. In general, I am extremely enthusiastic about the publication of this paper.

R: We are delighted that the Reviewer #1 found our paper exciting and of value for researchers in the field and for the broader research community. We also thank the Reviewer #1 for the valuable and constructive comments to help improve the manuscript.

However, I have found a few issues that I think should be addressed in advance of final publication which I describe below:

*1) An important part of the authors' model is that Dilp8, which is expressed in the epidermis in response to 20-hydroxyecdysone at puparium formation, signals to Lgr3-expressing neurons in the CNS to control pupariation behaviors. The authors explain that they were not able to generate a tissue-specific dilp8 knockout or knockdown to test this directly. However, they argue that their experiment rescuing AR in a dilp8 mutant by using a ubiquitously-expressed Dilp8 (*tub>dilp8*) expressed after MIT, supports their model (Fig. 2I). While it does show that dilp8 expression after MIT is sufficient to rescue AR, this experiment doesn't address the important spatial part of their model. I would suggest that the authors do a similar experiment and see if this rescue can happen using an epidermal driver to express Dilp8 specifically in the epidermal cells. This rescue would then tell us that the Dilp8 expressed in the epidermis can actually rescue AR, through communicating with the 18A01 neurons. This would also be useful for their conclusion that this same signaling pathway, from epidermis to 18A01 neurons, is required for GSB in figure 4H.*

R: We agree with Reviewer #1 that our rescue experiment only addressed the temporal requirement of *dilp8* in the pupariation behaviors, not the spatial requirement. However, the reason why we have not attempted a spatial rescue is because of the hormonal nature of the Dilp8 factor, i.e., a spatial rescue could be consistent with a model, but it would not be definitive proof of the requirement of the factor in that tissue. Nevertheless, we have successfully addressed this question more directly by revisiting the tissue-specific loss-of-function experiments with a new RNAi line + double-epidermal-GAL4-driver strategy. We have included the results of these experiments in Supplementary Fig. 1F, G; Fig. 3D, E. and have made the necessary modifications in the text (in blue) and in the material and methods sections.

Briefly, in our manuscript we show that Dilp8 activity needs to be very strongly and/or completely removed, as by the CRISPR-Cas9 nulls or KO alleles described in our manuscript, in order to see strong aspect ratio effects (Fig. 1 and Supplementary Fig. 1 and new panels therein). Knowing this, we hypothesized that if we expressed *dilp8* RNAi in the epidermis using a combination of two strong epidermal GAL4 lines (new Supplementary Fig. 2G), we could silence *dilp8* expression to such a level that we would see aspect ratio and maybe even GSB defects. To do so, we placed the two strongest epidermal GAL4 drivers that we had (A58-GAL4 and Eip71CD-GAL4) together with a new, stronger, *dilp8* RNAi construct (*dilp8*-IR-TRIP) and a UAS-Dicer construct to potentiate the RNAi effect. Whereas neither GAL4 driver alone affected aspect ratio or GSB when expressing *dilp8*-IR-TRIP +/- UAS-Dicer, when both drivers were used together, we found a statistically-significant increase in puparium aspect ratio (new Fig. 3E) and a statistically-significant fraction of animals failed to perform GSB (new Fig. 5I), whereas all controls behaved as expected. Hence, Dilp8 levels in the epidermis is a key factor for proper puparium aspect ratio and GSB. We believe these data, together with the additional experiments described below make a strong case that the proposed model is correct.

This text was inserted in the subsection (new text underlined):

“The Dilp8-Lgr3 pathway is required for puparium morphogenesis”

“Similar phenotypes were observed in five *dilp8* loss-of-function mutants generated here by CRISPR/Cas9-mediated directed mutagenesis^{48, 49} (Fig. 1E, F, Supplementary Fig. 1A), upon ubiquitous RNAi knockdown of *dilp8* (*dilp8*-IR^{TRIP}, see also Methods and Supplementary Fig. 1B-G), and in an independent knock-out allele *dilp8*^{KO} (Ref.⁴⁰, and Supplementary Fig. 1H, I).”

and in the second paragraph of the subsection:

“Dilp8 is required during pupariation for proper puparium morphogenesis”

“To genetically test for the spatial requirement of *dilp8* in the epidermis, we genetically knocked-down *dilp8* using the epidermal drivers A58> and Eip71CD> (A58>*dilp8*-IR^{TRIP} and Eip71CD>*dilp8*-IR^{TRIP}) and quantified puparium AR. However, neither condition altered the AR when compared to control genotypes (Fig. 3D, E). Attempts to use tissue-specific knockout of *dilp8* using a UAS-driven CRISPR-Cas9 system were unfortunately unsuccessful due to epistatic epidermal phenotypes caused by Cas9 expression (see Methods and Supplementary Fig. 3A, B). As puparium morphogenesis was particularly sensitive to *dilp8* levels and incomplete loss or silencing of *dilp8* expression leads to normal puparium formation (Supplementary Fig. 1B-G), we hypothesized that in order to observe the *dilp8* knockout AR phenotype using the RNAi strategy, we would have to increase the strength of the RNAi in the epidermis. To do this, we combined the

epidermal GAL4 drivers together (*A58+Eip71CD>dilp8-IR^{TRIP}*). As expected, knockdown of *dilp8* using the combined drivers significantly increase puparium AR when compared to each control genotype (Fig. 3D, E). We conclude that epidermis-derived *dilp8* is required for proper puparium morphogenesis.”

2) The characterization of PMP is quite interesting and potentially valuable. However, for it to be most useful for researchers who want to follow up on these experiments, it needs to be clearer how the authors are defining the beginning and end of each period. For instance, even in WT animals (figure 3C) it is not obvious how the transition from post-GSB1 to post-GSB2 is defined. The authors say that they are “characterized by different total *mhc>>*GCaMP-fluorescence fluctuation patterns” (page 15, line 13), but I couldn't find where the authors had clearly defined these different patterns or how they used them to assign the boundary. It would be useful for the authors to give us a more complete description of how those transition points are set. This becomes especially important when interpreting mutant phenotypes.

R: We agree with Reviewer #1 that the GCaMP fluctuation transition points should be more clearly defined. To address this we have added the definitions in the text and in the Methods sections. We also added the new Supplementary Fig. 6G,H (post-GSB effects of *Lgr3* mutation and post-GSB1-2 transition criteria), the new Supplementary Fig. 11 (*mhc>>*GCaMP-fluorescence WT profile examples), and the new Supplementary Fig. 12a-e (*mhc>>*GCaMP-fluorescence *Lgr3* mutant profile examples).

Below we exemplify the modifications:

Transition from post-GSB1 to post-GSB2 (this text was added into the third paragraph of the section subtitled “*dilp8* and *Lgr3* mutants do not perform GSB or post-GSB”:

“The transition from one stage to the other is smooth, without a clear limit between them, and both types of contractions (post-GSB1 and 2) can coexist during the transition. Thus, we arbitrarily established a boundary between stages after the occurrence of typically one or two *mhc>>*GCaMP-fluorescence peaks that were longer than preceding and subsequent ones. These contractions hence clearly mark the end of post-GSB1 and the beginning of post-GSB2 (Supplementary Fig. 6g, h).”

For instance, how are the authors determining the duration of pre-GSB in *dilp8* mutants (figure 3J)? Why are those patterns that they highlight in green clearly pre-GSB in nature?

R: We have added this description to a new Methods subsection called “Criteria to call pre-GSB *mhc>>*GCaMP-fluorescence peaks”

“Criteria to call pre-GSB *mhc>>*GCaMP-fluorescence peaks in mutant animals

All mutant animals show peaks of GCaMP fluorescence once they stop wandering. These peaks can result from peristaltic waves or whole-body contractions and were considered to be pre-GSB-like contractions only after confirming that the peak corresponded to a contraction of the body. This is certainly a conservative criteria, as those peaks that were not considered to be pre-GSB-like contractions because they were weak by visual inspection could have actually been part of the pre-GSB subprogram.

In mutant animals, the beginning of pre-GSB is determined by the first short-pre-GSB contraction that can be observed, as defined above. Mutant animals do not progress beyond short-pre-GSB stage. Then, the end of pre-GSB is reached when *mhc*>>GCaMP-fluorescence peaks do not lead to body shortening, which coincides with cuticle sclerotization. Examples of representative GCaMP traces of mutant animals that perform pre-GSB contractions or that lack identifiable generalized body contractions after completion of wandering stage are provided (Supplementary Fig. 12).”

As regards the problem of determining the duration of pre-GSB in the *dilp8* mutant, please see the new Supplementary Fig. 12. We also underscore the fact that we call them pre-GSB-short or pre-GSB-like, since, apart from the facts described above, we have no way to be 100% sure they are “clearly pre-GSB in nature”.

Alternatively, looking at the effects of mDopa on *dilp8* mutants (figure S5 D and E), it isn't clear how those stage assignments are being made. How is the beginning of pre-GSB defined? the end?

R: This was also answered in the new text pasted above. Namely, in mutant animals, the beginning of pre-GSB is determined by the first pre-GSB-short contraction that we can observe. This corresponds to a peak in *mhc*>>GCaMP fluorescence signal which is associated with a generalized body contraction. Mutant animals do not progress beyond pre-GSB-short stage. Then, the end of pre-GSB is therefore reached when *mhc*>>GCaMP fluorescence peaks do not lead to body shortening, which coincides with cuticle sclerotization.

Another difficulty that arises from the characterization of PMP is that it is unclear what the phenotypic variation is within experimental replicates. For the *dilp8* mutants, the authors show us two different traces (presumably two different animals, figure 3J and K) that look extremely different. Do those represent the range of phenotypes? are they representative samples? Is this variability seen in WT controls as well?

R: Yes. While some *dilp8* mutants perform only pre-GSB-like contractions, others fail completely in performing PMP. The fraction of the animals that fail had been quantified (Fig. 4E and Supplementary Fig. 4E). To make this clearer we have also added other representative figures to further illustrate the phenotypic variation (new Supplementary Fig. 12). This variability is also seen in *Lgr3* mutants, but is NOT seen in either WT controls, as shown in the respective quantifications of the samples (Figures 4 and Supplementary Figs. S4-6) and new examples provided (new Supplementary Fig. 11).

What about other mutants? It would be valuable if the authors could provide some information that could help readers appreciate the ranges of variation, both for evaluating the findings as well as to inform future experiments who would like to extend these studies.

R: We agree with Reviewer #1. We have provided a variety of *mhc*>>GCaMP-fluorescence profiles as examples in the new Supplementary Fig. 11-12.

3) The mDopa experiments have some issues with interpretation, and I think this part of the model is the weakest. For figure 5E, the authors demonstrate that mDopa treatment reduces puparium AR in *dilp8* mutants. They interpret this result as demonstrating that *dilp8* mutants don't achieve proper AR due to excess sclerotization produced by loss of *dilp8* signaling. However, the changes produced in AR by mDopa are the same in *dilp8* mutants and *dilp8*⁺ controls. The correct interpretation of this

experiment that the effects on AR from *dilp8* and mDopa appear function completely independently. Another way to put this is that the change in AR phenotype produced by loss of *dilp8* or mDopa treatment produce exactly the same change regardless of the other perturbation. This would suggest that mDopa and *dilp8* alter AR through independent mechanisms. The *Lgr3* data (figure 5F) is more compelling that there may be an interaction between *lgr3* and mDopa, but the effect of the *lgr3* mutant on AR is much stronger as well. It seems like the interpretation suggested by the data is that *lgr3* may have a *dilp8*-independent effect on Dopa-mediated sclerotization. I think the authors should re-evaluate this data, address these interpretations and reflect them accordingly in their model.

R: We partially agree with Reviewer #1. We agree that the mDopa experiments are weak on their own, but we are not comfortable revising the model, as argued briefly below. The fact that mDopa treatment of *dilp8* or *Lgr3* mutants does not bring the puparium aspect ratio (AR) values to the extra-low AR values found in mDopa-treated WT puparia shows that the mutant animals either have increased sclerotization and/or lack the behaviors that reduce the AR. Our experiments show that both of these are true. The key finding for the former is described in these sentences about mDopa treatment of *dilp8* and *Lgr3* mutants: “*a-Methyl*dopa treatment strongly increased the number of detectable pre-GSB contractions (Supplementary Fig. 5f) and mildly reduced their period (Supplementary Fig. 5g). This demonstrates that cuticle sclerotization negatively affects puparium AR by antagonizing pre-GSB number and frequency”. Hence, mDopa treatment affects *dilp8* and *Lgr3* mutant *mhc*>>GCaMP-fluorescence profiles, whereas they look normal in WT animals. Even though we go on to show that *dilp8* and *Lgr3* also achieve further AR reduction by an additional mechanism -the pre-GSB short-to-long switch- we are left with no option than to accept that precocious cuticle sclerotization is also a part of the *dilp8* and *Lgr3* phenotype. This is indeed consistent with the other independent findings we report, such as the *Tub*>*dilp8* temporal-rescue experiments.

4) Interpretations of GSB with the *R18A01* driver are difficult and should be avoided. For instance, in 6I it is not clear whether the loss of GSB is due to the intersectional expression of *Lgr3*-IR, or the presence of the *R18A01* driver.

R: We agree with Reviewer #1, and we state this in the text in the section “*Lgr3* is required in *R18A01* neurons for GSB”(new text is underlined)

“The fact that the GSB rescue is incomplete could suggest that *Lgr3* is an exquisitely limiting factor in the presence of *R18A01*>. Alternatively, a second factor in the *R18A01*> line could affect GSB but not AR, in an *Lgr3*-independent manner. For these reasons, conclusions on GSB based on the *R18A01*> driver should be taken cautiously.”

We consider that it is important to keep our cautious interpretations of these experiments as described in the text, as they are transparent and should hopefully help the readers understand why we avoid to conclude on aspects of GSB based solely on experiments with the *R18A01* drivers alone. As we mention in the text, in the case of experiment “Fig. 7i”, it is impossible to conclude whether or not the loss of GSB is due to the intersectional expression of *Lgr3*-IR or to the presence of the *R18A01*>> driver. This is why experiments, including the *Lgr3* genetic rescue, with the *R48H10*> driver, which has no detectable effect on puparium AR or GSB on its own, were so critical for our conclusions regarding the role of *Lgr3* in the 6 VNC neurons. Please see also the new discussion (2nd paragraph of Discussion section).

Minor issues:

5) An initial characterization of the Lgr3 Cis-regulatory regions would be valuable. Through much of the first half of the paper the authors describe the “two separate populations of neurons marked by R19B09 and R18A01” (Page 4 line 22-23) regulate larval duration and puparium morphogenesis separately, but didn't show these distinct populations (in Fig. 6 they show the R48H10 expression pattern, which seems to also influence puparium morphogenesis. It is also important to acknowledge that subsets of regulatory regions may also produce neomorphic patterns of expression, as regulatory enhancers may be separated from regulatory suppressors.

R: We agree with Reviewer #1 and have added images showing the expression patterns of each driver (R19B09> and R18A01>) to Supplementary Fig. 2j, k. The complete confocal stacks for each CRM are also available at <https://flweb.janelia.org/cgi-bin/flew.cgi>. Please also notice that we do show an image of R18A01> expression pattern in the thoracic region of the VNC in the Supplementary Fig. 9a, depicting the partial overlap with neurons expressing endogenous sfGFP::Lgr3.

We also agree and are completely aware of the fact that the ectopic placement of these subsets of regulatory regions can lead to neomorphic patterns of expression - they do! - but we argue that this is irrelevant to the usage we make of them as tools for driving expression in subsets of neurons. The key experiments here are the overlap of the GAL4 expression pattern with actual endogenous sfGFP::Lgr3 coupled with the Lgr3-IR loss-of-function (RNAi) experiments, which together show that Lgr3 is actually expressed and required in some of these neurons. For instance, the intersection of the R18A01 and R48H10 CRMs overlap with only 6 VNC neurons that actually express detectable endogenous sfGFP::Lgr3.

6) Not really clear to me how the authors conclude that “dilp8 is a direct target of 20HE-dependent signaling in the carcass”. (page 7 lines 20-21). Is this based on the timing of the result? are there EcR binding sites in Dilp8? Or are they saying that Dilp8 responds to ecdysone signaling and this could either be through direct binding of EcR to the Dilp8 regulatory regions, or through indirect transcriptional regulation? I would request that the authors clarify their interpretation of this experiment.

R: We thank the Reviewer #1 for spotting this lack of clarity. We have clarified this interpretation by adding the following sentence to the text to the 2nd paragraph of section:

“20HE signaling induces dilp8 transcription in the cuticle epidermis during pupariation”

... “These results suggest that *dilp8* is a direct or indirect target of 20HE in the larval carcass. The timing of the *dilp8* transcriptional response to 20HE are consistent with a model where *dilp8* is a direct target of very late 20HE-dependent signaling, probably the strongest and last peak preceding pupariation (at -4 h), whereas *pale* is induced by smaller and earlier 20HE peaks, probably the midthird-instar transition peak, which is linked to the initiation of salivary glue protein production in the salivary gland^{18, 34}.”

and after describing the new EcR-IR experiments, we now conclude (4th paragraph, same section):

“Even though we have not assayed for direct binding of EcR to the *dilp8* locus, the results described above are consistent with a cell-autonomous, direct regulation of *dilp8* by the *EcR*.”

Point-by-point response to Reviewers' comments.
Heredia et al. NCOMMS-20-39967-T

7) The authors describe how Dilp8/Lgr3 signaling regulate a “thresholded system” or a “thresholded morphogenetic mechanism” (Page 10, line 5 and page 12, lines 6-10). It is not clear to me what the authors exactly mean by this and it would be helpful if they could explain this concept more clearly.

R: We thank the Reviewer #1 for spotting this lack of clarity. We have included the following sentence in the end of the section “*Proper anterior retraction requires the Dilp8-Lgr3 pathway and is essential for survival*” to make our explanation more clearer:

... “...the mechanism might involve the setting of a threshold that defines a yes or no response (e.g., proper anterior retraction or not) to an intensifying morphogenetic process.”

8) For the evaluation of pre-PMP locomotion patterns in Lgr3 mutants (extended data figure 3), it is not clear that there are enough data points for each condition to evaluate the differences between the conditions. More datapoints would be helpful here.

R: We agree with Reviewer #1. We have conducted more experiments and added more datapoints in Fig. 5c, d, e (new Lgr3 data), Supplementary Fig. 4e, f, g, h (new Lgr3 background data), and Supplementary Fig. 5c, d (new Lgr3 background data). The results are all consistent with previous interpretations.

9) On page 10, lines 28-30 are confusing. Replace “posteriorly to” with “after” or “following” and insert “likely occur during” in place of “likely”

R: We have done these modifications.

10) Page 15, line 28: “later” should be “latter”

R: We have changed it as suggested.

11) Figure 5 legend (page 18, lines 15 and 16), J and K legends are mislabelled

R: We have corrected the figure legends.

12) For Figure 5J it is not clear whether the mRNA was isolated from the whole animal or just the carcass.

R: We have modified the text as follows: “To test if the dilp8 transcripts are upregulated before T0 in a more precise manner, we obtained samples from whole animals exactly 5 min after they had performed GSB”

In summary, I think this will be a very valuable paper to the field. After the authors address the issues detailed above, I would strongly support its publication in Nature Communications.

R: We thank the Reviewer #1 for the insightful, careful, and very helpful review of our manuscript. We hope that we have addressed all points raised either with new data or with a suitable explanation and hope the Reviewer #1 is satisfied and can support the publication of our manuscript in Nature Communications.

Reviewer #2 (Remarks to the Author):

In this article, Heredia et al investigated the role of the relaxin-like Dilp8-Lgr3 pathway in puparium morphogenesis and further characterized pupariating behavior by describing the sequence of events taking place during pupariation motor program (PMP). By combining long-term live imaging and quantitative image analyses with very sophisticated genetic experiments, the authors found that ecdysone-dependent Dilp8 expression in epidermal cells signals to a cluster of 6 Lgr3-positive interneurons located in the VNC to promote PMP progression.

This is a well-designed and executed study that provides novel insights into the role of inter-organ crosstalk in controlling an innate behavior. The article is very well written, yet some description of the different PMP behavioral subunits were quite long and confusing. Nevertheless, I have only two main points to strengthen the key findings, and some other minor issues.

R: We are delighted that the Reviewer #2 considered our study well-designed, executed, and insightful. We also thank the Reviewer #2 for the valuable and constructive comments to help improve the manuscript. We understand the criticism regarding the description of the different PMP behavioral subunits, but as other Reviewers requested further descriptions of the PMP behavioral subunits, we would be happy to discuss with the editor to try to find a compromise (we can move parts of the descriptions to a suitable Supplementary Information section, at the editor's discretion).

Mayor points,

1- Dilp8, produced by damaged imaginal discs, has been shown to inhibit ecdysone biosynthesis, and increased levels of 20-hydroxyecdysone in late L3 larvae is known to trigger pupariation. Thus to strengthen the proposal that Dilp8 is required downstream of 20HE to promote puparium morphogenesis, it would be important to show that ecdysone biosynthesis and/or signaling is not affected when interfering with dilp8/Lgr3 pathway.

R: We thank the Reviewer #2 for raising this important point. To address it we have performed new experiments described in the new Figure 7N and in the new section "*Lgr3 activity in pupariation-controlling neurons do not affect ecdysone biosynthesis or activity*"

Briefly, we have performed quantitative RT-PCR with synchronized T0 animals of animals lacking *Lgr3* activity in the R48H10 neurons (*R48H10>Lgr3-IR*), which have aberrant aspect ratio (AR) and do not perform GSB, (*R48H10>+* and *UAS-Lgr3-IR/+* served as controls) and measured the relative mRNA levels of the ecdysone biosynthesis genes *phantom* (*phm*) and *disembodied* (*dib*) and the ecdysone-responsive gene, *E74B*. Consistent with our hypothesis, we did not find any statistically-significant difference in either gene or pathway (new Figure 7N). Importantly, the *R48H10>Lgr3-IR* condition was specifically chosen to avoid any confusion from the R18A01> insertion genotype and any interference with the timing of the onset of pupariation which occur when using *dilp8* or *Lgr3* mutations [there is a 4-h anticipation of pupariation in those genotypes (Colombani et al., Science 2012 and Garelli et al., Nat Commun 2015)].

2- It would be important to fully demonstrate the source of Dilp8 is the epidermis, and to circumvent technical limitations in knocking-down dilp8 expression specifically in epidermal cells, the authors could use da>EcRRNAi to suppress Dilp8 expression in these cells (as shown in Extended Data Figure 2C) and measure AR.

This experiment would also answer whether ecdysone signaling in epidermal cells is required for proper pupariation.

R: We agree with the Reviewer #2. To address this point, which was also partially raised by Reviewer #1, we carried out the experiments described above in major point 1 of the Reviewer #1, and two additional experiments as described below.

The results are described in the third paragraph of subsection “20HE signaling induces dilp8 transcription in the cuticle epidermis during pupariation”.

“To genetically test if *dilp8* transcription in the epidermis occurs downstream of 20HE signaling, we knocked-down the *ecdysone receptor (EcR)* gene with RNAi (*UAS-EcR-IR*) in the epidermis using two epidermal GAL4 lines *A58-GAL4* and *Eip71CD-GAL4* (*A58>EcR-IR* and *Eip71CD>EcR-IR*; see Supplementary Fig. 2g) and quantified puparium AR and *dilp8* mRNA levels by qRT-PCR in synchronized wandering stage (108 h after egg laying) and WPP T0 stage animals. The *UAS-EcR-IR* transgene alone (*EcR-IR/+*) and *EcR* knockdown in the fat body using the *pumpless-GAL4* line (*ppl>EcR-IR*) served as a negative controls for epidermal expression (Supplementary Fig. 2g). Results showed a statistically-significant increase in puparium AR in *A58>EcR-IR* and *Eip71CD>EcR-IR* animals, but not in all other controls (Fig. 2e, f). Furthermore, as expected, we observed a statistically-significant decrease in *dilp8* mRNA levels in *A58>EcR-IR* and *Eip71CD>EcR-IR* WPP T0 animals, but not in all other controls (Fig. 2g). We conclude that epidermal *EcR*, but not fat body *EcR*, is critical for the achievement of peak *dilp8* mRNA levels in WPP T0 animals. Interestingly, the puparium AR increase produced by *EcR* knockdown in the epidermis was much stronger than what we observed in *dilp8* or *Lgr3* animals (compare Fig. 1a-f with Fig. 2e, f). This is consistent with a scenario where ecdysone signaling regulates additional aspects required for proper puparium morphogenesis, apart from *dilp8* transcription, such as cuticle sclerotization. Accordingly, the cuticle of *A58>EcR-IR* animals partially or completely fails to sclerotize, whereas cuticle sclerotization is apparently complete in *dilp8* or *Lgr3* mutants). In line with this rationale, a fraction of the control *ppl>EcR-IR* animals had defective anterior retraction (Fig. 2e, lower panels), which could suggest a role for the fat body (or any other *ppl>*-positive tissue) in this process. We nevertheless hypothesize this is unlikely to be related to the expression of *dilp8* in the epidermis, as *ppl>* does not drive significant expression in cuticle epidermal cells at this developmental stage (Supplementary Fig. 2g).

The fact that *A58>EcR-IR* and *Eip71CD>EcR-IR* WPP T0 animals are so severely affected and that the *dilp8* mRNA peak is so sharp in time, can cast doubt on the precision of the samples collected, despite our efforts to avoid this problem by carefully monitoring each animal and establishing criteria for as WPP T0 as wandering behavior cessation, spiracle extrusion, and body contraction cessation. To circumvent this limitation, we dissected the carcass of staged, 96-h *A58>EcR-IR* larvae and quantified *dilp8* mRNA levels following incubation with 20HE or vehicle for 6 h *ex vivo*, as performed above. Carcasses from control animals carrying the *EcR-IR* transgene alone or with *EcR*

knockdown in the fat body (*ppl>EcR-IR*) served as controls. Results showed that whereas 20HE strongly induced *dilp8* in *EcR-IR/+* or *ppl>EcR-IR* animals, there was no statistically-significant induction of *dilp8* by 20HE in the carcasses of *A58>EcR-IR* animals (Fig. 2h). Even though we have not assayed for direct binding of EcR to the *dilp8* locus, the results described above are consistent with a cell-autonomous, direct regulation of *dilp8* by the *EcR*.”

and in the third paragraph of subsection “*Dilp8* is required in the epidermis for GSB”:

“As the genetic knockdown of EcR in the epidermis (*A58>EcR-IR* or *Eip71CD>EcR-IR*) significantly reduced *dilp8* mRNA levels, we also assayed for GSB in these animals. However, knockdown of EcR in the epidermis did not interfere with GSB (Supplementary Fig. 7a). This is consistent with our findings that neither genotype completely eliminated *dilp8* transcript levels (Fig. 2g), and is in line with the model where the epidermally-derived Dilp8 is required downstream of ecdysone-signaling for proper GSB.”

An important finding here is that knockdown of EcR-IR in the epidermis did not affect the execution of GSB per se (new Supplementary Fig. 7A). As EcR knockdown in the epidermis using either *A58>* or *Eip71CD>* drivers did not completely abrogate *dilp8* mRNA and we have shown that complete *dilp8* elimination is required to observe the *dilp8*-dependent effect on GSB, we conclude that EcR-IR in the epidermis does not affect GSB because there is still enough *dilp8* to induce the pre-GSB to GSB transition in these animals. These results are all consistent with our final model where 20HE signaling induces cuticle sclerotization program, the *dilp8* mRNA-induction program, and the body contraction program. As expected, we show that the cuticle sclerotization and *dilp8* mRNA program are a result of EcR signaling in the cuticle epidermis.

These results are consistent with the experiments described above and are also consistent with independent results posted in a BioRxiv preprint by the P. Leopold laboratory, who also demonstrate that the WPP-T0 *dilp8*-mRNA peak is Ecdysone-dependent (Boulan et al., BioRxiv 2020).

3- What are the consequences of pupariation defects for animal fitness or survival? For instance, *dilp8ag51* and *ag55* show similar AR defects (Fig. 1F) but very different lethality (Extended Fig. 2K). The authors should include a more direct discussion on this issue.

R: We agree with the Reviewer #2. To address this point, we have added the following sentence (new text underlined) to the text and a new set of data to Supplementary Fig. 3M, showing that there is no consistent correlation between AR and survival:

“To test if this lethality was linked to puparium AR defects we measured AR of puparia from animals that eclosed or not. Only one out of four *dilp8* mutant genotypes surveyed showed a statistically significant difference between the puparium AR of animals that survived or died (Supplementary Fig. 3i). Hence, we conclude that there is no consistent association between survival and puparium AR. To test if this lethality was linked to anterior retraction defects, we followed pupal viability in animals with gross anterior retraction defects. We find that 100% of animals with visible anterior retraction defects fail to eclose, suggesting that proper anterior retraction is critical for pupal viability (Fig. 3f). Furthermore, ~50% of animals without clear anterior retraction defects also die. It is likely

that those animals still have subtle anterior retraction defects (for example, they could be unable to seal the cuticle after retraction of the mouth hooks). Nevertheless, the fact that a fraction of mutants achieves WT-level puparium AR, at least something that looks like proper anterior retraction of the pre-spiracular segments, and survives proves that the Dilp8-Lgr3 pathway is not *per se* the signal for anterior retraction.”

Minor points,

- Pag. 9, line 20: Subheading: “Proper anterior retraction requires the Dilp8-Lgr3 pathway and is essential for survival” would be more appropriate?

R: We have modified the text accordingly.

- Pag. 10, line 4: “the peak can be detected as soon as 1.0 h after WPP T0 (“T30”, Fig. 2A)”. I think it should say “T60”.

R: We have modified the text accordingly.

- Pag. 11, line 16-18: “After 2-4 s, large ventral intersegmental (longitudinal) muscles..., a process that helps extrude the anterior spiracles (Fig. 3I, Supplementary Video 2).” I don’t think this is the proper video.

R: We have modified the video reference accordingly.

- Fig. 2A: miss-alignment between letters and pupal images.

R: We have modified the figure accordingly.

- Fig. 4G: the authors show that R18A01>Lgr3 rescues AR, but only partially rescues GSB in Lgr3ag1 animals. Does it mean that GSB is not required for proper puparium morphogenesis (AR)?.

R: Yes. GSB is the glue-spreading behavior and is not *per se* required for proper puparium aspect ratio (AR)-determination. This is because it occurs after the major AR-determining behavior: pre-GSB. Proper execution of the behaviors that follow GSB -the “Post-GSB-1 and -2 behaviors”- might also slightly contribute to AR. We currently believe this occurs via the active (muscle-contraction-dependent) maintenance of puparium AR while full cuticle sclerotization takes hold.

- Fig. 5B: why the effect of Dilp8 overexpression was analyzed in a dilp8 mutant background? I would have anticipated to see this experiment in wild-type animals.

R: We agree with the Reviewer #2. The explanation is simple, however: as it was critical to express *dilp8* after the midthird instar transition to avoid delayed pupariation, and we had the temporally-controlled setup ready for the rescue experiments (with Tub-GAL80), we made good use of it. Our other *dilp8* overexpression setups were not temporarily-controlled.

- Extended Data Figure 1A: what do red boxes represent?

R: They represent the six conserved cysteines of the insulin-like peptide superfamily of peptides. We have added an appropriate description in the respective figure legend.

Point-by-point response to Reviewers' comments.
Heredia et al. NCOMMS-20-39967-T

- Extended Data Figure 2G: UAS-dilp8gRNA should be changed (as far as I understand it is not a UAS line).

R: This is indeed a UAS-line made from the plasmid: "pCFD6" (please see Port and Bullock, Nature Methods, 2016). From their website: <http://www.crisprflydesign.org/plasmids/> "pCFD6: UAS-t::gRNA: Plasmid for tissue-specific gRNA expression with the Gal4/UAS system. Can express one or multiple tRNA-flanked gRNAs."

We thank again the Reviewer #2 for the valuable and constructive comments to help improve the manuscript. We hope to have addressed all of the points raised with either new data or with a suitable explanation and hope the Reviewer #2 is satisfied and can support the publication of our manuscript in Nature Communications.

Reviewer #3 (Remarks to the Author):

This manuscript outlines an impressive study that demonstrates that dilp8, a Drosophila relaxin-like peptide, plays a key role in the formation of the puparium, which is necessary for the insect to complete metamorphosis. Not only is this an elegant genetic study that outlines the role of dilp8 in puparium formation, the role of the dilp8 receptor in this same process, but the authors beautifully characterise the behavioural processes involved in the formation of the puparium. Importantly, the authors go the pains of narrowing down the cells responsible for the behaviours involved in puparium formation. This manuscript represents an enormous amount of work, and I think the authors are to be commended on the quality and rigour of their experimental paradigms. I have a few relatively minor comments that I feel would improve the readability of this manuscript.

R: We are very happy that the Reviewer #3 found our work elegant, comprehensive, and rigorous. We also thank the Reviewer #3 for the valuable and constructive comments to help improve the manuscript, which we address below.

In section 10, page 5, I'm not sure what you mean when you say you can manipulate imaginal disc derived Dilp8 using the cis-regulatory module R19B09? What is this construct? Is this part of the regulatory region of Lgr3, or is it just a construct that is expressed in the same neurons as Lgr3? Figure 1I suggests this construct contains regulatory elements specific to lgr3, presumably the source of Dilp8 is not relevant at the level of an enhancer element. I think this section needs some careful rephrasing to avoid confusion.

R: We are sorry for the confusion. In the manuscript we state that: "Imaginal disc-derived Dilp8 acts on a subpopulation of *Lgr3*-positive CNS neurons that can be genetically manipulated using the cis-regulatory module (CRM) *R19B09* 25–28, 50, 52, 53 (Fig. 1H-I)." To avoid further confusion we have rewritten this paragraph as follows (new text is underlined):

"Imaginal disc-derived Dilp8 acts on a subpopulation of *Lgr3*-positive CNS neurons that can be genetically manipulated using the cis-regulatory module *R19B09*²⁵⁻²⁸ (Fig. 1h-i and Supplementary Fig. 1j), which consists of the ~3.6-kb 7th intron of the *Lgr3* locus^{50, 52, 53}. *R19B09*-positive cells include a bilateral pair of neurons, the *pars intercerebralis* *Lgr3*-positive (PIL)/growth coordinating *Lgr3* (GCL) neurons, which respond to Dilp8 by increasing cAMP levels, and are thus considered the major candidate neurons to sense the Dilp8 imaginal tissue growth signal^{25-27, 46}. We reasoned

Point-by-point response to Reviewers' comments.
Heredia et al. NCOMMS-20-39967-T

that if the neurons that require *Lgr3* to inhibit ecdysone biosynthesis upon imaginal tissue stress are the same neurons that require *Lgr3* to control puparium morphogenesis, then knockdown of *Lgr3* in *R19B09*-positive cells, but not in the other *Lgr3* CRM-positive cells, should increase puparium AR."

In figure 2G, it might be worth adding "exogenous Dilp8" to the diagram of the dilp8 ^{-/-} larva in which you overexpress dilp8.

R: We have modified the figure accordingly.

Section 25, page 7: What is the MIT ecdysone peak? Are you talking about the peak of ecdysone that induces glue production (Sgs3) in the salivary glands?

R: Yes, as defined by Warren et al., Dev Dyn 2006 and Hackney et al., PLoS One 2012. To clarify, we have added the following sentence (underlined) to the end of the 2nd paragraph of section "20HE signaling induces dilp8 transcription in the cuticle epidermis during pupariation":

... "whereas *pale* is induced by smaller and earlier 20HE peaks, probably the midthird-instar transition peak, which is linked to the initiation of salivary glue protein production in the salivary gland".

The plots in figure 4 are misleading. If your Y axis is % of animals performing a behaviour, why have a Yes/No legend for a binary trait? Isn't it enough to see that 0% of the dilp8^{-/-} and lgr3^{-/-} animals perform the behaviour? Also, I'm not seeing the "same blue letters" indicated in the legend or the binomial statistics.

R: We have changed the graphs as requested and have added asterisks to indicate the results of the statistical analyses.

I didn't understand the mDOPA experiments in section 20-30 on page 19. I think it would be helpful to explain the difference between sclerotization and melanization, and how you expect each to affect the behaviours associated with pupariation, here.

We have expanded the following explanation (new text underlined) to the third paragraph of the section: "*The Dilp8-Lgr3 pathway antagonizes cuticle sclerotization during PMP*":

" α -Methyldopa inhibits the enzyme Dopa decarboxylase (Ddc), which converts DOPA to dopamine in the epidermis, an essential step in insect cuticle sclerotization^{68, 69} (Fig. 6c). α -Methyldopa treatment is thus expected to have at least two effects: to inhibit cuticle sclerotization by reducing the amount of available Dopamine that gets fed into the cuticle sclerotization pathways, and a strong melanization of the cuticle, as the unconverted excess of the Dopamine precursor, DOPA, becomes available to the alternative black-melanin production pathway (Fig. 6c). Cuticle melanization per se

is not expected to interfere with pupariation. As expected, α -methyl dopa treatment led to strong melanization of the cuticle, confirming that Ddc was efficiently inhibited (Fig. 6c, d).”

I'm not so convinced of the role of dilp8's role in heterochrony. Yes, it's true that heterochrony is defined as a change in the timing of the progression of development, but the heterochronic genes in *C. elegans* cause cells to skip whole stage or else remain stuck in the cell divisions of a single stage. It seems to me that dilp8 is instead important in the correct implementation of a developmental transition. This is decidedly different from skipping a stage (or failing to progress beyond a life stage).

R: We understand the point made by the Reviewer #3, but we also understand the definition of heterochrony as being broader and going beyond that defined by the *Caenorhabditis elegans* heterochronic genes (as described and discussed by Carl Thummel in *Dev Cell*, 2001 and by Eric Moss in *Current Biology*, 2007; see for instance, the concept of “temporal boundary and temporal identity”), hence it includes the mechanisms of timing of the implementation of developmental transitions. This definition also implies that skipping a stage or failing to progress beyond a life stage are the most drastic forms of developmental heterochrony. It also implies that hormones such as Ecdysone and Thyroid Hormones are heterochronic factors as well. Please notice that the Dilp8/Lgr3 pathway would encompass even the most conservative definition of skipping a life stage: animals lacking dilp8 or Lgr3 skip the GSB stage, and are “locked” (temporarily) in the pre-GSB-short stage of development. As GSB is an innate behavior, it is a bona-fide part or stage of the developmental program of *Drosophila melanogaster*.

Minor comments:

- Are the AR, CRM, MIT, PMP, and GSB acronyms necessary? I find they disrupt the flow of the text because I have to remind myself what they mean.

R: To address this concern, we have removed the CRM and MIT acronyms in the text (we only kept them in the Figures, with the definitions in the Fig legends), but maintained AR, PMP, and GSB in the text, as they are used too many times and the PMP and GSB acronyms are used to define the other stages of the pupariation motor program such as pre-PMP or pre/post-GSB. We also removed the AEL (after egg laying) abbreviation in the text. We hope that this compromise is satisfactory.

- To me, carcass implies the remains of dead animals. I wonder if body wall or integument and body wall muscle would be more accurate?

R: We have included the definition of “integument and body wall muscle” in certain places (as in Fig. 2 and Supplementary Fig. 2 legends) as a further description of the “carcass” term. However, we found it important to keep the term “carcass”, as it is the term used in previous defining studies (e.g., Chintapalli et al., 2007, *Nat. Genet.*, and Andres & Thummel, *Methods in Cell Biology*, 1994), one of which are used as a source of tissue atlas gene expression information in the FlyBase database/website, providing a reference for comparison with previous studies.

We thank again the Reviewer #3 for the valuable and constructive comments to help improve the manuscript. We hope to have addressed all of the points raised with either new data or with a suitable explanation and hope the Reviewer #3 is satisfied and can support the publication of our manuscript in *Nature Communications*.

Reviewer #1 (Remarks to the Author):

The authors have made substantial additions and changes to the manuscript and have added important experiments that bolster their model. In particular, the ability to determine that Dilp8 function in the epidermis is necessary for AR and GSB is a key finding.

I think for my point 3, the authors have successfully argued that while the individual mDopa treatment experiment has some ambiguity in its interpretation, the preponderance of the additional evidence supports their model.

I'm fully satisfied with the revisions made by the authors in this paper. I would reiterate that it is a valuable and interesting study that will have a meaningful impact on our field and I would fully support its publication.

Reviewer #2 (Remarks to the Author):

The authors have addressed the issues raised in the original submission.

Reviewer #3 (Remarks to the Author):

The authors have addressed all of my comments.

MS: NCOMMS-20-39967A

Heredia *et al.*

Point-by-point response to reviewers' comments:

Authors' responses are highlighted in blue.

REVIEWERS' COMMENTS

Reviewer #1 (Remarks to the Author):

The authors have made substantial additions and changes to the manuscript and have added important experiments that bolster their model. In particular, the ability to determine that Dilp8 function in the epidermis is necessary for AR and GSB is a key finding.

I think for my point 3, the authors have successfully argued that while the individual mDopa treatment experiment has some ambiguity in its interpretation, the preponderance of the additional evidence supports their model.

I'm fully satisfied with the revisions made by the authors in this paper. I would reiterate that it is a valuable and interesting study that will have a meaningful impact on our field and I would fully support its publication.

Reviewer #2 (Remarks to the Author):

The authors have addressed the issues raised in the original submission.

Reviewer #3 (Remarks to the Author):

The authors have addressed all of my comments.

R: We are happy that the reviewers agreed that all comments were addressed. We thank the reviewers for their help in improving the manuscript.